Corrected: Publisher Correction

# Transcriptomic alterations during ageing reflect the shift from cancer to degenerative diseases in the elderly

Peer Aramillo Irizar[1], Sascha Schäuble[2,3], Daniela Esser[1], Marco Groth [3,4], Christiane Frahm[3,5], Steffen Priebe[3,6], Mario Baumgart[3,7], Nils Hartmann[3,8], Shiva Marthandan[3,9], Uwe Menzel[3,6], Jule Müller[5], Silvio Schmidt[3,5], Volker Ast[10,11], Amke Caliebe[12], Rainer König[10,11], Michael Krawczak[12], Michael Ristow [3,13], Stefan Schuster[3,14], Alessandro Cellerino[3,7,15], Stephan Diekmann[3,16], Christoph Englert[3,8,17], Peter Hemmerich[3,9], Jürgen Sühnel[3,18], Reinhard Guthke[3,6], Otto W. Witte[3,5], Matthias Platzer[3,4], Eytan Ruppin[19] & Christoph Kaleta [1,3]

Disease epidemiology during ageing shows a transition from cancer to degenerative chronic disorders as dominant contributors to mortality in the old. Nevertheless, it has remained unclear to what extent molecular signatures of ageing reflect this phenomenon. Here we report on the identification of a conserved transcriptomic signature of ageing based on gene expression data from four vertebrate species across four tissues. We find that ageing-associated transcriptomic changes follow trajectories similar to the transcriptional alterations observed in degenerative ageing diseases but are in opposite direction to the transcriptomic alterations observed in cancer. We confirm the existence of a similar antagonism on the genomic level, where a majority of shared risk alleles which increase the risk of cancer decrease the risk of chronic degenerative disorders and vice versa. These results reveal a fundamental trade-off between cancer and degenerative ageing diseases that sheds light on the pronounced shift in their epidemiology during ageing.

[1] Research Group Medical Systems Biology, Institute of Experimental Medicine, Christian-Albrechts-University Kiel, D-24105 Kiel, Germany. [2] Jena University Language and Information Engineering Lab, Friedrich-Schiller-University Jena, D-07743 Jena, Germany. [3] GerontoSys JenAge Consortium, D-07745 Jena, Germany. [4] Genome Analysis Lab, Leibniz Institute on Aging–Fritz-Lipmann-Institute, D-07745 Jena, Germany. [5] Hans Berger Department of Neurology, Jena University Hospital, D-07747 Jena, Germany. [6] Systems Biology and Bioinformatics Group, Leibniz Institute for Natural Product Research and Infection Biology–Hans-Knöll-Institute, D-07745 Jena, Germany. [7] Biology of Ageing Lab, Leibniz Institute on Aging–Fritz-Lipmann-Institute, D-07745 Jena, Germany. [8] Molecular Genetics Lab, Leibniz Institute on Aging–Fritz-Lipmann-Institute, D-07745 Jena, Germany. [9] Imageing Facility, Leibniz Institute on Aging–Fritz-Lipmann-Institute, D-07745 Jena, Germany. [10] Integrated Research and Treatment Center, Center for Sepsis Control and Care (CSCC), Jena University Hospital, D-07747 Jena, Germany. [11] Network Modeling, Leibniz Institute for Natural Product Research and Infection Biology–Hans Knöll Institute, D-07745 Jena, Germany. [12] Institute for Medical Informatics and Statistics, Christian-Albrechts-University Kiel, D-24105 Kiel, Germany. [13] Energy Metabolism Laboratory, Swiss Federal Institute of Technology (ETH) Zurich, Schwerzenbach/Zürich, CH-8603, Switzerland. [14] Department of Bioinformatics, Friedrich-Schiller-University Jena, D-07743 Jena, Germany. [15] Laboratory of Neurobiology, Scuola Normale Superiore, University of Pisa, I-56100 Pisa, Italy. [16] Molecular Biology Lab, Leibniz Institute on Aging–Fritz-Lipmann-Institute, D-07745 Jena, Germany. [17] Faculty of Biology and Pharmacy, Friedrich-Schiller-University Jena, D-07743 Jena, Germany. [18] Biocomputing Lab, Leibniz Institute on Aging–Fritz-Lipmann-Institute, D-07745 Jena, Germany. [19] Department of Computer Science and Center for Bioinformatics and Computational Biology, University of Maryland, College Park, MD 20742, USA. Peer Aramillo Irizar, Sascha Schäuble and Daniela Esser contributed equally to this work. Correspondence and requests for materials should be addressed to C.K. (email: c.kaleta@iem.uni-kiel.de)

The process of ageing has been studied across many diverse species and is recognized as the dominant risk factor for most human diseases[1,2]. There is a remarkable conservation of ageing-associated pathologies such as cancer, cardiovascular disease, as well as cognitive decline across vertebrates,[3–5] and a strong association between life-history traits and gene expression across the mammalian tree of life[6]. Despite these similarities, previous studies that have investigated transcriptomic signatures of ageing have identified only few genes that show a conserved transcriptional regulation across species[7–11]. In consequence, even though a number of ageing-associated transcriptomic changes have been linked to lifespan[10,12,13], the relationship between ageing-associated transcriptional changes as well as ageing-associated pathologies has remained largely unclear[8,14]. This is quite surprising as it has been shown that human diseases including ageing-associated pathologies are often associated with specific transcriptional signatures and that reverting these gene expression states back to their original healthy patterns can successfully point to potential treatments[15–17].

One particular reason for this discrepancy might be the prevailing view that ageing-associated changes necessarily have to promote ageing-associated diseases such as cancer, cardiovascular diseases, neurodegenerative diseases and metabolic diseases. Thus, when investigating the influence of ageing-associated changes on ageing diseases it is often not taken into account that ageing might have different influences on specific ageing disorders[8,18,19]. Indeed, as reported previously in the context of cancer[20], epidemiological data shows pronounced shifts in the cause of mortality between different subtypes of diseases during ageing (Fig. 1a). In particular, the contribution of cancer to all-cause mortality is highest among the 60 year olds, and then steadily decreases while the contribution of degenerative disorders gradually increases (Fig. 1a). This inverse pattern manifests also when examining the incidence of these disorders (measuring the number of new diagnoses of a disease relative to population size). Here the incidence of degenerative ageing-associated diseases keeps rising monotonically to the oldest old while cancer incidence peaks in the 75–84 year olds and then tapers off (Fig. 1b, cf. ref. 20). Furthermore, malignant transformation rates, which measure the rate at which pre-malignant cells progress through the steps of carcinogenesis, grow most rapidly at the age of 50 and decline across all cancer types above the age of 70 years (Fig. 1c, Supplementary Note 1). Similar observations characterize ageing-associated cancer mortality in mice (Supplementary Note 1). Pathologically, these ageing-associated changes in cancer epidemiology are reflected by slower growth, a reduced metastatic potential and a generally reduced life-threatening potential of cancer in the oldest old[21]. Different explanations for these observations, including the antagonistic role of some tumor suppressive mechanisms on degenerative ageing disorders and antagonisms between individual disorders, have been discussed[20,22–25]. However, the underlying causes for the marked difference in ageing-associated epidemiology between cancer and degenerative diseases in the old have remained unclear[20,22].

On the basis of these observations, we hypothesized that transcriptomic alterations may account for the shift from cancer to degenerative diseases in the late stages of ageing. To address this hypothesis, we generate a comprehensive transcriptomic data set of ageing covering four tissues in four different vertebrate model organisms. Analyzing this data, we find that the ageing transcriptome is associated with a shift toward the expression signatures of chronic degenerative diseases (cardiovascular, metabolic and neurodegenerative disorders) while it shifts away from cancer-associated gene expression signatures. Considering genomic risk variants associated with susceptibility to ageing-associated diseases, we find a strikingly similar antagonistic trend, whereby shared risk alleles between cancer and degenerative disorders antagonistically predispose to either type of disease while protecting from the other. These results provide a clear link between ageing-associated transcriptomic changes as well as ageing diseases and reveal a fundamental, conserved trade-off between the molecular changes occurring in cancer and degenerative disorders of ageing.

## Results

**A conserved functional signature of transcriptomic ageing.** To investigate the hypothesis that ageing-associated transcriptomic alterations are linked to pronounced shifts in the epidemiology of ageing-associated diseases, we analyzed a cross-species ageing transcriptomic data set covering 531 samples including five ageing time points (Supplementary Note 2 and Supplementary Fig. 1) and four tissues (blood, brain, liver and skin) in four model organisms of ageing: humans, mice, the zebrafish *Danio rerio* and the short-lived killifish *Nothobranchius furzeri* (Supplementary Fig. 1 and Supplementary Note 2). All samples (of which 297 are first presented in this study) were generated on the same platform following standardized protocols (Methods). Apart from mouse, human and zebrafish as most important vertebrate model organisms of ageing, we included data from *N. furzeri* which is the shortest-lived vertebrate with an average lifespan of three months for the shortest-lived strain[26] and shows a high concordance in its ageing phenotypes with other vertebrates[5].

Previous studies have reported only little overlap in ageing-associated differential expression of individual genes between species[7,8]. Thus, to obtain a transcriptomic signature of ageing, we assessed ageing-associated changes on an aggregate, functional level by determining conserved differentially regulated processes (Methods). Processes were derived across three different ontologies: Gene Ontology[27], KEGG Pathway[28] and a genome-scale reconstruction of human metabolism[29] (Methods section). As the number of differentially regulated processes across species increases steadily with age for all ontologies considered (Supplementary Fig. 2), we chose the first and the two last sampled time points (cf. Supplementary Note 2) for deriving a transcriptomic signature of ageing. Using unbalanced type-II analysis of variance, we find 171 of 900 processes that are significantly differentially regulated using Gene Ontology, 18 of 74 processes for KEGG Pathways and 10 of 43 human metabolic pathways (Fig. 2, Methods section). Most of the processes differentially regulated between individual age groups were also differentially regulated in the comparison between the young and the two old age groups (Methods section). This functional ageing signature shows a high concordance with previously reported ageing-associated transcriptional changes in individual species and tissues: we find an induction of the immune system, indicative of the low-grade inflammation in the elderly[30], downregulation of cell cycle-associated processes[31] and downregulation of many developmental as well as cell differentiation pathways[32]. In addition, we detected an upregulation of glycan degradation pathways, the downregulation of co-factor-associated metabolic processes and a pronounced induction of several signaling pathways. In difference from previous studies[7,8], we observe a strong conservation of ageing-associated changes on a functional level both across species (including humans) as well as across tissues (Supplementary Fig. 3 and Supplementary Note 3), also on the level of individual genes (Supplementary Note 3).

**Association with cancer and degenerative diseases.** To quantify the similarity between ageing and disease-associated transcriptomic data we defined an ageing-mediated disease alignment

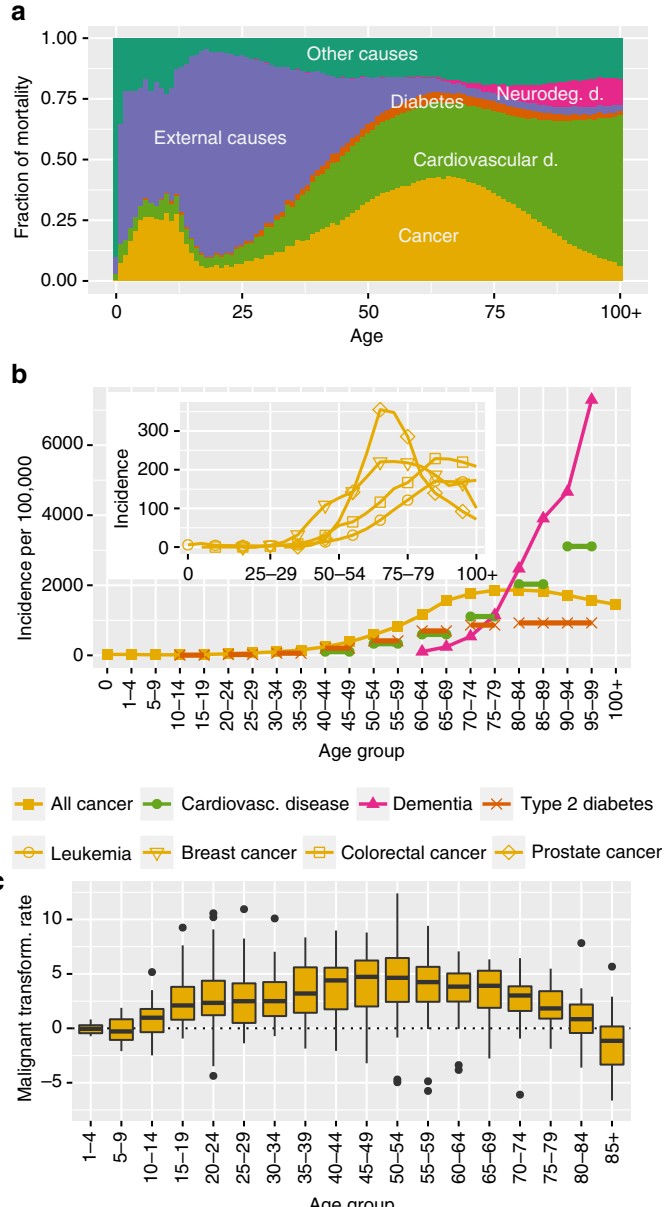

**Fig. 1** Epidemiology of ageing-associated diseases. **a** Contribution of different disease categories to total human mortality (Supplementary Note 1). **b** First diagnosis (incidence) of ageing-associated diseases and several cancer types (inset) across age groups. Lines across several age groups indicate cases in which only data combining several age groups was available (Supplementary Note 1). **c** Malignant transformation rates across 34 types of cancer across age groups. The center line corresponds to the median, box limits to 25th and 75th percentiles and whiskers to 1.5 times the interquartile range

score (AMDA score, Methods section). The AMDA score is positive if ageing aligns the transcriptome with that of a given disease, that is, ageing is associated with similar changes in the transcriptome to that observed in a disease vs. healthy controls (called disease signature alignment). The AMDA is negative if ageing moves the transcriptome away from a disease signature, that is, ageing is associated with transcriptomic changes that are opposite to those observed in disease vs. healthy controls (called disease signature reversion). Please note that the AMDA by its definition measures the shift toward or away from a disease signature and hence does not imply that a particular tissue has a specific disease state.

We determined AMDA scores for 22 ageing-associated data sets and 30 data sets of different human ageing-associated diseases. The 22 ageing data sets include the data sets that we have generated and reviewed above, tissue-specific ageing

signatures and previously reported human age-regulated gene sets (Supplementary Note 2). Hence, we did not only include gene sets based on differentially regulated processes but also on the basis of differentially expressed genes in the individual species and tissues. The 30 disease data sets include 14 different cancer data sets, 7 cardiovascular diseases data sets, 5 data sets of neurodegenerative diseases and 4 type 2 diabetes data sets (Supplementary Note 2). For cancer, data sets originates from affected tissue while for degenerative diseases data mostly originates from blood (Fig. 3) or affected tissue in several cases.

Computing the AMDA scores using these data we find that the expression signature of old individuals is shifted toward the signature of patients with cardiovascular diseases (including hypertension and atherosclerosis), neurodegenerative diseases (including mild cognitive impairment) and type 2 diabetes (including insulin resistance), compared to young individuals.

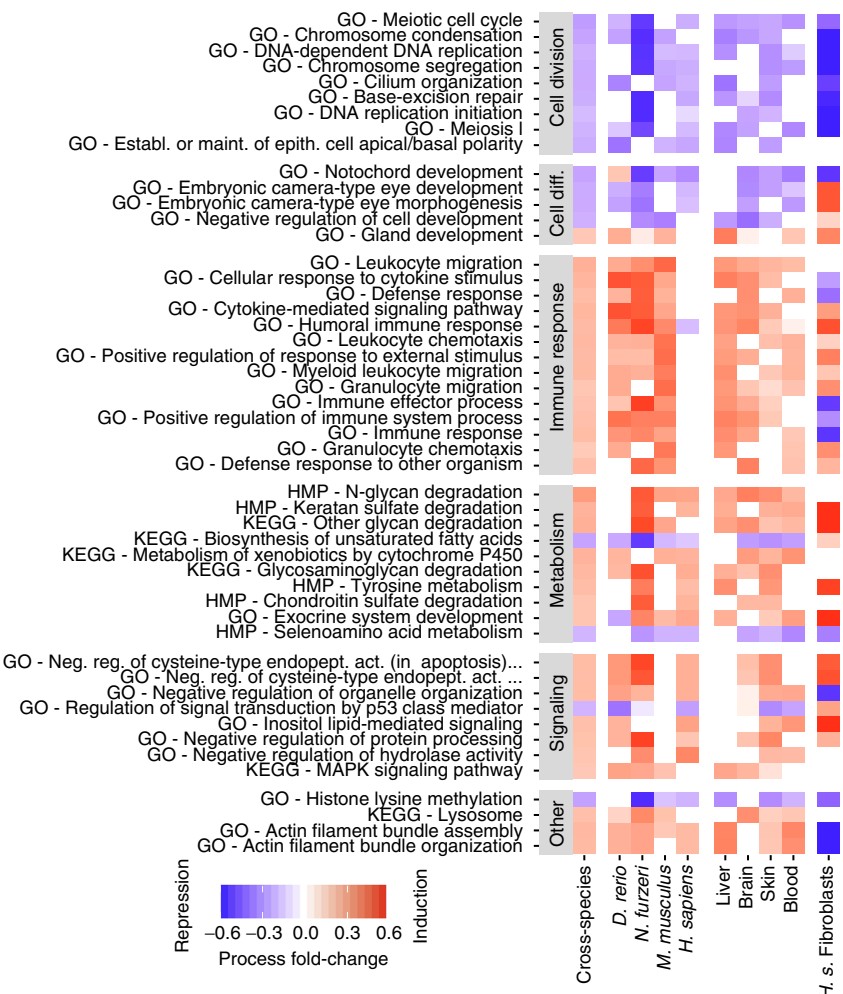

**Fig. 2** Ageing-associated transcriptional changes across species and tissues. The top 50 differentially expressed processes between young and old individuals in the cross-species comparison are shown across all three ontologies (indicated in front of the labels). The first column displays the main processes altered across all samples followed by columns presenting the results for individual species and tissues across species (only significantly ageing-regulated processes are displayed). Processes are grouped into functional categories and ordered in each category separately according to the p-value of ageing (see Methods). Abbreviations: HMP human metabolic pathways, *H.s. Homo sapiens*, GO gene ontology, KEGG Kyoto Encyclopedia of Genes and Genome

However, ageing moves the transcriptome away from cancer-associated gene expression signatures (Fig. 3a). Examining the transcriptomic changes occurring in cancer driver genes, oncogenes are enriched with ageing-repressed genes while tumor suppressors as well as genes that are frequently lost during tumorigenesis are enriched with ageing-induced genes (Supplementary Note 4).

Cellular senescence has been demonstrated to contribute to ageing-associated pathologies[33] and senescent cells have been reported to accumulate with age in many tissues across species[34,35]. We, therefore, tested to which extent cellular senescence contributes to the observed shifts in the ageing transcriptome relative to ageing diseases. We found that after removing gene sets from the analysis that are differentially expressed in senescing cell cultures or belong to proliferation-associated processes, we still observe a strong alignment of the ageing transcriptome with the expression signatures of degenerative ageing diseases and a reversal of the expression signature of cancer (Supplementary Fig. 4).

We further tested whether ageing-mediated disease alignment scores also reflect changes in cancer epidemiology in the middle age. This age group from 30 to 45 years is associated with

considerable changes in human reproductive biology[36,37] and shows a marked increase in malignant transformation rates (Fig. 1). For these age groups, in contrast to later age groups, we observed a strong alignment of the transcriptome with cancer in agreement with an acceleration of cancer incidence observed from malignant transformation rates (Supplementary Note 5).

**Comparison of longitudinal and cohort ageing**. Next, we performed a longitudinal analysis of ageing-mediated disease alignment at the individual's level, asking how they are associated with lifespan. To this end, we analyzed mice and *N. furzeri* ageing transcriptomic data for which samples were obtained at two time points of ageing from the same individual (Methods section). As in the cross-sectional analysis, the observed longitudinal ageing-associated changes occur in an opposite trend to gene expression signatures of cancer and are aligned with the expression signature of degenerative diseases (Fig. 3a), involving similar processes (Supplementary Note 6). We obtain similar results when considering subsets of age groups in our data for which the influence of cohort effects (changes in gene expression between age groups due to more susceptible individuals dying first) is mitigated

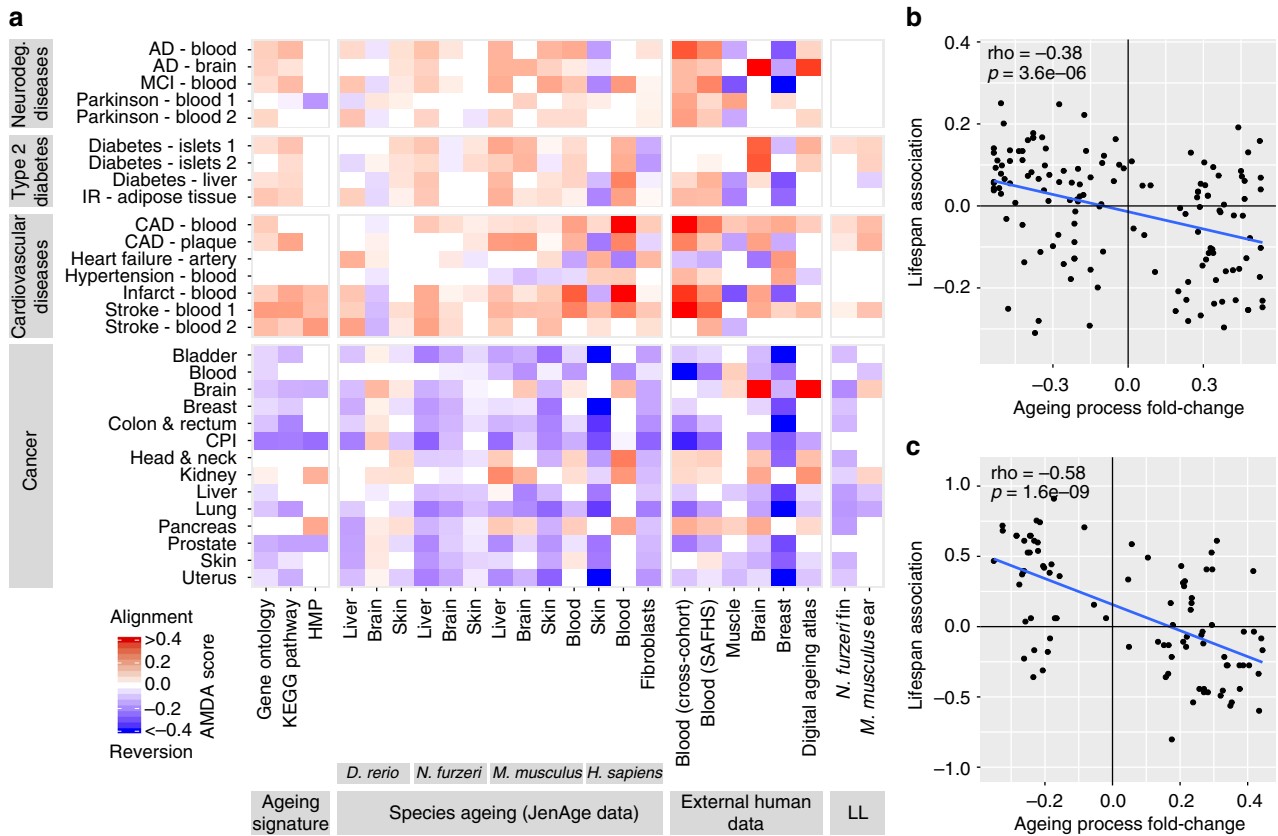

**Fig. 3** Association of ageing-associated gene expression changes with ageing diseases and lifespan. **a** Ageing-mediated disease alignment scores (AMDA scores) for different ageing (x axis) and disease data sets (y axis). Abbreviations: AD Alzheimer's disease, CAD coronary artery disease, CPI cancer proliferation index, cross-cohort cross-cohort study of blood ageing, HMP human metabolic pathways, IR insulin resistance, LL longitudinal ageing data, MCI mild cognitive impairment; SAFHS, San Antonio Family Heart Study. **b**, **c** Correlations between activity changes of processes during ageing (x axis) and their lifespan-associations (y axis, Pearson correlation) for *N. furzeri* (**b**) and mouse (**c**). The corresponding test statistics are provided in Supplementary Data 1

(Supplementary Note 7). These results suggest that the observed ageing-associated transcriptomic changes occur as a part of the ageing process in individuals. Changes in population heterogeneity, which arise due to an earlier death of individuals that are more susceptible to specific diseases, thus appear to have only small impact on the transcriptomic signature which we have identified.

As expected, the activities of processes induced during ageing are negatively correlated with lifespan while the activity of processes repressed during ageing shows mostly positive correlations (Fig. 3b, c). Thus, despite the ageing-associated reversion of cancer gene expression signatures, there is an overall negative association with lifespan.

**Key drivers of ageing-mediated disease alignment**. To assess which of the ageing-regulated processes has the strongest impact on the alignment between ageing and disease signatures, we defined a disease alignment contribution (DAC) score. This score measures the contribution of genes belonging to a specific process to the overall alignment of disease-specific expression signatures with ageing (a positive score denotes a process whose genes shift the ageing signature toward the given disease and a negative score denotes an inverse trend, cf. Methods section and Supplementary Note 8). The top ranking DAC score processes show the same direction of changes in the chronic degenerative diseases surveyed but an opposite one in cancer (Fig. 4a, b). Thus, processes induced in cancer are downregulated in degenerative diseases and

vice versa. This antagonism is also visible in a multi-dimensional scaling plot including all ageing and disease data, where the degenerative disease samples cluster together with the ageing data while cancer samples form a separate cluster (Fig. 4c, d, Supplementary Note 8). The upregulation of immune-associated and cell cycle-associated processes most strongly contributes to ageing-mediated disease alignment, either when considering all or just the human ageing data (Fig. 4b). This is in concordance with the important role of chronic inflammatory processes in the genesis of cardiovascular diseases, neurodegenerative diseases and type 2 diabetes[2,30] that drive tissue decline due to a constant activation of the immune response[38]. In context of cancer, the immune system, though often attributed a dual role[39,40], is the principal systemic response against the development of cancer if cell-autonomous mechanisms for cancer prevention fail[41], also reflected by the often immunosuppressive microenvironment of tumors[42]. Among the ageing-repressed processes, cell cycle-related functions (Figs. 2 and 4) show the strongest contribution to ageing-mediated disease alignment (Fig. 4b). This concurs with the role of cellular senescence as an important contributor to ageing-associated pathologies[33]. Moreover, cellular senescence is an essential tumor suppressive mechanism even though some aspects of the senescent phenotype can also contribute to tumorigenesis in pre-malignant cells[33].

We sought to identify potential transcription factors that mediate the observed antagonism. To this end, we determined motifs that are enriched in the promoter regions of most consistently ageing-regulated genes using mSigDB[43]

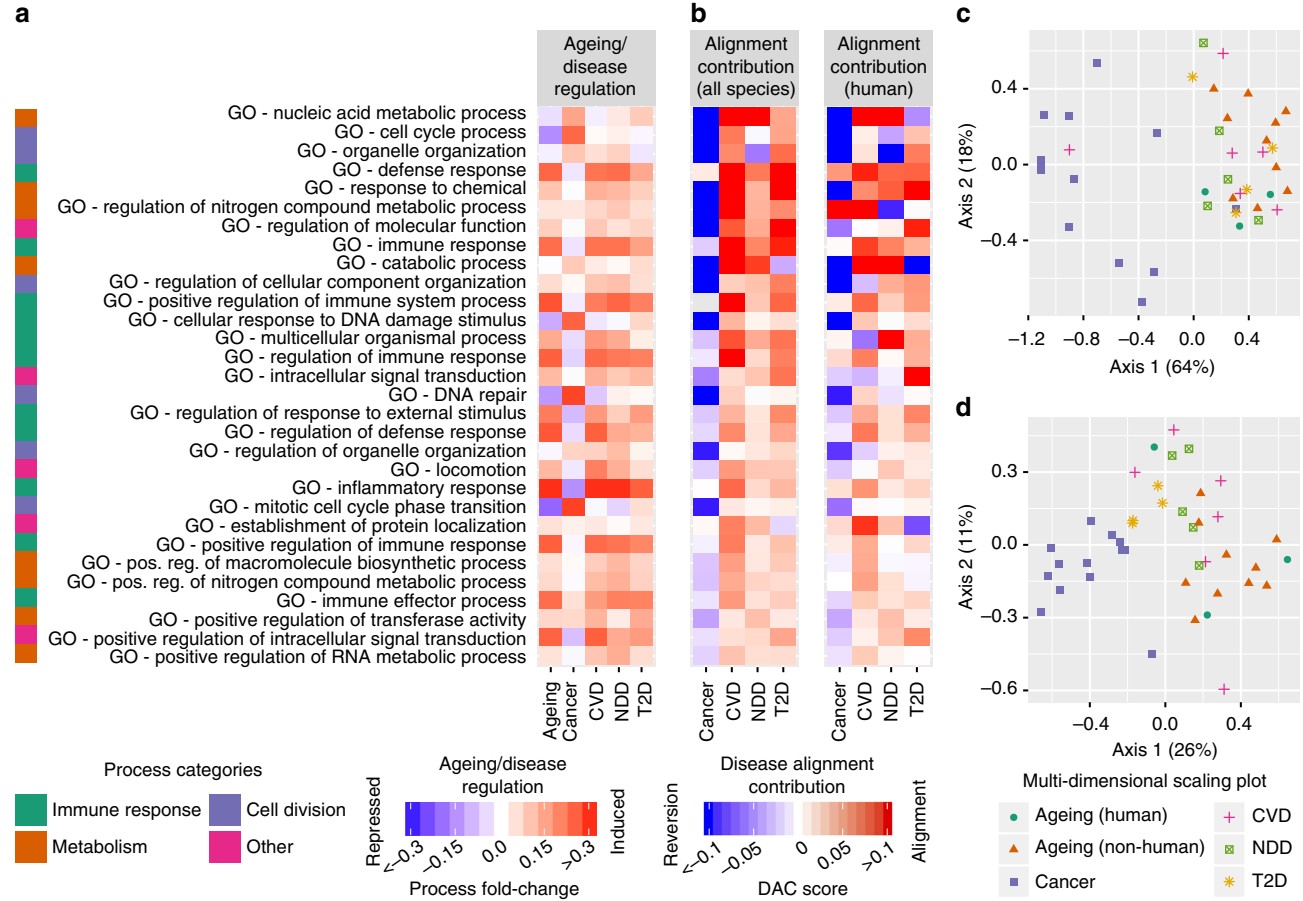

**Fig. 4** Role of ageing-regulated processes in disease signature alignment. **a** For each process, mean foldchanges of genes belonging to that process in the ageing and disease data sets are shown. Processes are sorted by average absolute DAC scores . **b** DAC scores of 30 top scoring processes across the four major disease categories for all ageing data (first column) and for human ageing data solely (second column). **c**, **d** Multi-dimensional scaling plot of disease and ageing-associated gene expression changes based on (**c**) process foldchanges and (**d**) gene expression foldchanges. Each point corresponds to a condition (disease or ageing data set); distances indicate similarity in gene expression changes between conditions. Numbers in brackets denote the variance explained by each axis. Abbreviations: CVD cardiovascular diseases, NDD neurodegenerative diseases, T2D type 2 diabetes

(Supplementary Note 2). We found that binding sites of the transcription factors E2F1, NF-AT, CEBPB, and AP1 showed the strongest enrichment among these genes. Intriguingly, all of these transcription factors are important regulators of both cancer as well as degenerative ageing diseases and are key regulators of immune as well as cell cycle processes[44–50]. This supports their central role in mediating the antagonism between cancer and degenerative ageing diseases.

**Antagonism between ageing diseases on the genetic level**. An intriguing quandary in the study of genetic determinants of human lifespan is the detection of yet only two genes that have been reproducibly associated with human longevity, ApoE and FOXO3A[51,52]. This is especially puzzling given the existence of hundreds of risk variants for ageing-associated diseases[53,54] and sizable study populations[51]. Indeed, the very old seem to possess a comparable number of genomic risk variants as younger controls[55]. Given these previous observations, we hypothesized that, analogous to the antagonism between cancer and degenerative ageing diseases on the transcriptomic level, there might exist antagonistic effects occurring on the genomic level that may at least partially contribute to this paucity of genomic findings associated with human longevity. Thus, risk alleles that predispose to cancer may actually protect from degenerative ageing diseases and vice versa. Through such antagonistic effects, the

influence of risk alleles on longevity would be reduced or even canceled out in the extreme case. In line with our hypothesis, for one of the two known lifespan-associated genes, ApoE[56], only one variant shows an impact on lifespan while two other variants show no influence on lifespan but are antagonistically associated with age of onset of cancer and cardiovascular diseases[25]. While previous systematic studies have reported that ageing-related processes are enriched for shared genes that are associated with different types of ageing diseases[18,19], they have failed to account for the direction of effects, that is, how alternative alleles at the same genomic location influence disease risks.

To test our hypothesis, we compiled a list of all known genomic single-nucleotide polymorphisms (SNPs) associated with human ageing-associated diseases from GWAS Catalog[57]. Subsequently, we identified risk SNPs shared between cancer and degenerative diseases and determined how alleles at these loci influenced disease risk, while accounting for shared heritability through linkage disequilibrium (Methods section). We defined synergistic risk SNPs as those for which the same allele predisposes to at least one type of cancer as well as one degenerative ageing disease and antagonistic risk SNPs as those where one allele predisposes to at least one type of cancer and the alternative allele to at least one degenerative ageing disease (Methods section). Thus, in the case of antagonistic risk SNPs, one allele predisposes to one type of disease while protecting from the other and vice versa.

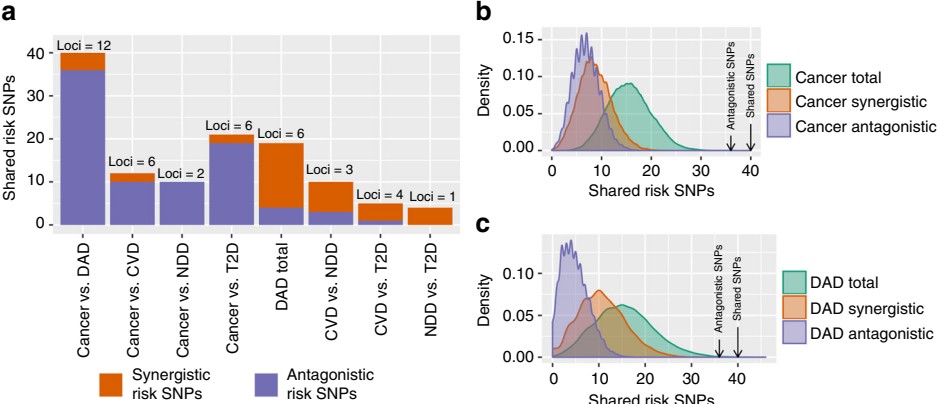

**Fig. 5** Antagonism between cancer and degenerative ageing diseases on the genetic level. **a** Shared synergistic and antagonistic risk SNPs between cancer and degenerative ageing diseases (DAD). "DAD total" indicates the total number of synergistic and antagonistic risk SNPs in the pairwise comparison between all three categories of degenerative ageing diseases. The total number of independent genomic loci in which the shared SNPs are contained is displayed on top of the bars. CVD cardiovascular diseases, NDD neurodegenerative diseases, T2D type 2 diabetes. **b** Randomization tests. Distributions depict the frequency at which the corresponding number of synergistic, antagonistic and total number of shared risk SNPs were encountered between cancer risk SNPs and randomly drawn risk SNPs from non-ageing-associated traits in 10,000 repetitions. Arrows indicate the actual observed number of antagonistic and total shared risk SNPs between cancer and degenerative ageing diseases. **c** Same as in **b** for degenerative ageing diseases instead of cancer

Overall, we identified 12 independent genomic loci containing 40 shared risk SNPs between cancer and degenerative ageing-associated diseases with 36 of them being antagonistic and only 4 being synergistic risk SNPs (Fig. 5a, Supplementary Data 1). This represents a highly significant bias toward antagonistic risk SNPs (binomial test $p$-values of $7.28 \times 10^{-15}$ and $1.99 \times 10^{-9}$, comparing the frequency of antagonistic risk SNPs for degenerative ageing diseases and cancer to non-ageing-associated traits, respectively, Supplementary Note 8). In contrast, comparing degenerative diseases among each other, we identified an opposite bias, composed of 15 synergistic and 4 antagonistic risk SNPs (Fig. 5a, Supplementary Note 8). More generally, we find that cancer as well as degenerative ageing diseases tend to have more synergistic than antagonistic risk SNPs with other non-ageing-associated traits contained in GWAS catalog (Fig. 5b, c). Furthermore, the overlap of risk SNPs between cancer and degenerative ageing diseases is much larger than expected by chance based on the overlap between both individual disease categories and non-ageing-associated traits reported in GWAS catalog (Fig. 5b, c, Supplementary Note 8).

The genomic locus with the largest number of shared risk SNPs contains the long non-coding RNA ANRIL that has been found to regulate several adjacent cell cycle regulators including one of the most important markers of cellular senescence and ageing, p16INK4a[58]. This locus encompasses the strongest known genomic risk variant for cardiovascular disease[59] as well as risk loci for neurodegenerative diseases[60] and type 2 diabetes[61]. Our analysis reveals that these risk SNPs are antagonistic to risk variants for several major cancers[62,63]. Another locus is contained within the coding region of SH2B3, also known as Lnk, which is an important component of inflammatory signaling[64] containing risk SNP predisposing to cardiovascular diseases and their risk factors[65]. The alternative allele at this genomic position has been reported as a risk SNP for several major cancers[62,66].

## Discussion
We find that ageing is associated with an antagonism between cancer and degenerative chronic diseases. It is important to emphasize that the trade-off that we have identified is distinct from other trade-offs considered in the context of the evolutionary theory of ageing such as the concept of antagonistic pleiotropy[67] or trade-offs identified in the context of the disposable soma theory[68]. The concept of antagonistic pleiotropy posits that ageing-associated changes might be driven by processes that are benefiting the survival to reproduction in the young while having deleterious effects in the old[67]. The disposable soma theory states that the rate of ageing is driven by a trade-off between the investment into reproduction and the maintenance of the individual[68]. Thus, antagonistic pleiotropy considers trade-offs in the effects of molecular processes between young and old individuals while the disposable soma theory considers trade-offs involving the ageing process as a whole. In contrast, we have identified a trade-off between cancer and degenerative diseases as integral part of the ageing process.

The antagonism between cancer and degenerative diseases is evident, though manifested differently, on both the transcriptomic and genomic level. On the transcriptomic level, this antagonism is reflected by a shift of the ageing transcriptome toward that of degenerative diseases and its divergence from cancer-associated gene expression signatures. Examining these alterations at the cellular process level, we find an inverse relation between the direction (up/downregulation) of transcriptomic changes occurring in key processes altered in cancer vs. degenerative ageing disorders. On the genomic level, this trade-off is reflected by the existence of a large number of risk alleles having opposite effects on the predisposition to cancer vs. degenerative ageing diseases. The latter may contribute to the observation that most genomic variants detected for ageing-associated diseases have no influence on human lifespan. As our analysis indicates, key contributors to ageing-mediated disease alignment are immune-associated as well as cell cycle-related processes, with the former being suppressed in cancer and induced in degenerative diseases and the latter induced in cancer and downregulated in degenerative ageing diseases. Since the immune system is the principal systemic barrier to cancer development[41] and there is an accumulation of potentially carcinogenic DNA damage with age[69,70], our results suggest that ageing-associated inflammatory processes that purport chronic diseases might actually be geared to counteract precancerous events. These results show that ageing-associated changes do not necessarily always promote ageing diseases but emphasize a separate consideration of cancer vs. degenerative disorders.

As the transcriptomic signatures of cancer and degenerative disorders antagonize each other in key cellular processes

involving the immune response and cell cycle regulation, it logically follows that the transcriptomic alterations occurring late in life may thus not be able to drive us away (or bring us closer) to the expression signatures of both cancer and degenerative disorders. Our analysis shows that the ageing transcriptome moves closer to that of the degenerative disorders, which is in agreement with their ever-increasing incidence. Finally, the conservation of the ageing response indicates that the trade-off between cancer degenerative ageing diseases might represent a fundamental characteristic of ageing in vertebrates.

In summary, our work provides a reference for conserved ageing-associated alterations on the transcriptional level and demonstrates a link of these alterations to ageing-associated pathologies. While we mostly report on associations and cannot infer causal relationships, these results represent an important basis for future functional studies that examine the mechanisms that mediate the antagonism between cancer and degenerative ageing diseases.

## Methods

**Zebrafish samples**. Zebrafish of the TüAB strain were kept in groups of 20–30 animals under standard husbandry conditions. For expression profiling, zebrafish were killed with MS-222 and cooled on crushed ice. Sampling was performed at noon/early afternoon to avoid effects of circadian rhythms. Brain, liver, and skin tissue from randomly selected male zebrafish at the age of 6, 12, 24, 36, and 42 months were dissected and stored in RNAlater (Qiagen, Hilden, Germany) at −80 °C. Total RNA was isolated with TRIzol (Life Technologies, Darmstadt, Germany) according to the instructions of the manufacturer. The protocols of animal maintenance and experiments were approved by the local authority in the State of Thuringia (Veterinär- und Lebensmittelüberwachungsamt, reference number J-SHK-2684-04-08/11).

***Nothobranchius furzeri* samples (cross-sectional/longitudinal)**. For expression profiling we obtained brain, liver and skin from randomly selected male animals of *N. furzeri* (strain MZM-04/10) from the corresponding cohorts at 5, 12, 20, 27, and 39 weeks of age. To avoid effects of circadian rhythms and feeding, animals were always sacrificed at 10 a.m. in fasted state. For tissue preparation, fish were killed with MS-222 and cooled on crushed ice. The protocols of animal maintenance and experiments were approved by the local authority in the State of Thuringia (Veterinär- und Lebensmittelüberwachungsamt, reference number J-SHK_22-2684-04-03-011/13).

**Mouse (cross-sectional/longitudinal) samples**. Male C57BL/6J mice at 2, 9, 15, 24, and 30 months of age were randomly selected from a breeding facility and deeply anaesthetized. Whole blood was taken by heart puncture and stabilized in PAXgene Blood RNA tubes according to manufacturer's instruction (PreAnalytiX GmbH, Switzerland). Furthermore, one cm2 hairless abdominal skin and 1 cm³ liver as well as one brain hemisphere were removed. For longitudinal sampling ears were clipped using a die cutter with a diameter of 3 mm (Natsume Seisakusho Co. Ltd., Japan) at 24 and 30 months of age. For the 30 months biopsy the second ear was clipped. All samples were immediately snap frozen and stored at −80 °C. All animal procedures were approved by the local government (Thueringer Landesamt, Bad Langensalza, Germany) and conformed to international guidelines on the ethical use of animals.

**Human samples**. Whole blood was taken from healthy male volunteers aged from 24 to 29, 45 to 50, 60 to 65, and 75 to 80 years and stabilized in PAXgene Blood RNA tubes. The same volunteers had a four mm punch biopsy performed above the knee inside the leg. The study was approved by the institutional review board of the Medical Faculty of the Friedrich-Schiller-University (registration number 3369-02/12). Informed consent was obtained from all subjects.

All human and murine samples were homogenized in QIAzol Lysis Reagent (Qiagen, Hilden, Germany) and 0.2 volumes chloroform were added. Following phase separation, the aqueous phase was transferred into a fresh tube, then 0.16 volumes NaAc (2 M, pH 4.0) and 1.1 volumes isopropanol were added. The RNA was precipitated by centrifugation and the pellet was washed with 75% ethanol. Total RNA was re-suspended in water and stored at −80°C until use.

**Human fibroblast cell cultures**. Primary human fibroblasts MRC-5, WI-38, BJ and IMR-90 were kept in groups obtained from ATCC (LGC Standards GmbH, Wesel, Germany). HFF cells were kind gifts of T. Stamminger (University Erlangen, Germany). All cell cultures were tested negative for mycoplasma contamination using DNA staining. Cells were cultured as recommended by ATCC in Dulbeccos modified Eagles low glucose medium (DMEM) with L-glutamine (PAA

Laboratories, Pasching, Austria), supplemented with 10% fetal bovine serum (FBS) (PAA Laboratories). Cells were grown under normal air conditions in a 9.5% $CO_2$ atmosphere at 37 °C. For sub-culturing, the remaining medium was discarded and cells were washed in 1 x PBS (pH 7.4) (PAA Laboratories) and detached using trypsin/EDTA (PAA Laboratories). Primary fibroblasts were sub-cultured in a 1:4 (=2 population doublings (PDs)) or 1:2 (=1 PD) ratio. For stock purposes, cryo-conservation of the cell strains at various PDs were undertaken in cryo-conserving medium (DMEM+10% FBS+5% DMSO). Cells were immediately frozen at −80 °C and stored for two to three days. Afterwards, cells were transferred to liquid nitrogen for long time storage. Re-freezing and re-thawing was not performed to avoid premature senescence. One vial of each of the five different fibroblast cell strains (MRC-5, HFF, BJ, WI-38 and IMR-90) was obtained and maintained in culture from an early PD. On obtaining enough stock on confluent growth of the fibroblasts in 75 cm² flasks, cells were sub-cultured into three separate 75 cm² flasks ("triplicates") and were passaged until they were senescent in culture.

**Purification of total RNA**. Total RNA was extracted as described before[71]. RNA extraction for all samples followed an a priori established standardized protocol. RNA quality and amount were determined using the Agilent Bioanalyzer 2100 with the RNA 6000 Nano Kit (Agilent Technologies). RNA integrity numbers varied between 5 and 10 depending on species and tissue. If RNA integrity numbers were below that threshold, samples were not used.

**RNA-seq**. In the case of blood samples, additional depletion of globin mRNA was applied using the GLOBINclear™ kit (Ambion, Thermo Fisher Scientific) following the manufacturer's protocol. Total RNA/globin mRNA-depleted RNA was introduced into either Illumina's TruSeq RNA sample preparation kit or TruSeq RNA sample preparation kit v2 following the manufacturer's description. The libraries were quality-checked and quantified using Agilent Bioanalyzer 2100 with the Agilent DNA 7500 kit (Agilent Technologies) and sequenced using either a HiSeq2000 or HiSeq2500 (Illumina) in high-output and single-end sequencing mode with a read length of 50nt. Sequence information was extracted in FastQ format using CASAVA v1.7/v1.8/v1.8.2 or bcl2fastq v1.8.3 or v1.8.4 (Illumina). Sequencing resulted in around 40–60 million reads per sample. Information about individual samples can be found in Supplementary Note 2 and Supplementary Data 1, sheet "Sample overview".

**RNA-seq analysis**. Read mapping against the respective species-specific reference genome (D. rerio: Zv9.73, M. musculus: GRCm38.69, H. sapiens: GRCh37.66) was performed using the Tophat splice alignment tool, version 2.0.6[72]. *N. furzeri* data was processed as described before[73,74]. Depending on species and tissue, 60–95% of reads could be mapped uniquely to the corresponding reference genome. To assign sequence reads to annotated transcripts, we processed the resulting BAM alignment files with the HTSeq software package[75], later with the featureCounts script[75]. For both methods, we made use of species-specific GTF gene annotation files that were downloaded from the Ensembl website[76]. Read counts per gene were further processed using the R software suite: Read counts were normalized with respect to individual transcript size using exon lengths provided by featureCounts and total amount of all mappable reads (library size), resulting in RPKM values (reads per kilobase of transcript per million reads mapped) for each gene. The counts and RPKM values served as raw data for all subsequent analyses. Outliers were detected using multi-dimensional scaling plots and not considered for further analyses (see Supplementary Data 1). For each organism, genes that showed no detectable expression (RPKM = 0) in one or more samples were excluded from down-stream analysis.

**Derivation of process ontologies**. To allow a functional comparison of gene expression data across tissue and species, we mapped expression values to process activity data based upon three different ontologies: Gene Ontology[27], KEGG Pathway,[28] and a metabolism-based ontology. Genes for each species were associated to processes using R packages biomaRt[77] for Ensembl's[76] Gene Ontology annotations and gage[78] for KEGG Pathway. For the metabolism-based ontology, we used the genome-scale human metabolic network, Recon 2.04, as a template[29]. Genes were mapped to metabolic subsystems according to the gene-reaction association of the model. For deriving metabolic functions, human genes were mapped to their respective homologs in the other three species using Biomart version 0.7[77].

For each ontology, processes were filtered such that only those processes annotated with at least five genes with measured expression in each species and tissue remained. The activity of a process in a sample was calculated as the sum of the expression values of all genes belonging to that process. Please note that this approach is equivalent to considering the average expression of all genes constituting a process due to the subsequent normalization steps described in the following. To render process activity for a specific ontology comparable across samples, process activities were first quantile normalized across all samples. Subsequently, the activity of each process was rank-normalized for each tissue in each species separately and ranks were scaled to a minimum of zero and a maximum of one. In total, we obtained process activity for 1563 processes for Gene Ontology, for 135 processes for KEGG Pathway and for 66 processes for Human Metabolic Pathways.

**Identification of significantly differentially regulated processes**. To detect processes that are differentially regulated with age, unbalanced type-II analysis of variance (ANOVA) was applied as implement in the 'car'[79] package of R. The outcome (dependent variable) was 'process activity' and apart from 'age group', 'species' (*M. musculus*, *H. sapiens*, *D. rerio*, and *N. furzeri*) as well as 'tissue' (blood, brain, liver, and skin) were treated as fixed independent factors. In the initial analysis, the factor 'age group' consisted of only two groups. From the five groups (young, $mature_1$, $mature_2$, $old_1$, $old_2$, Supplementary Note 2) all individual pairs of age groups were considered. After identification of the highest number of differentially regulated processes between the youngest and the two oldest time points of organismal ageing (Supplementary Fig. 2), the three age groups young, $old_1$ and $old_2$ were considered as factors in the analysis. All interactions up to three-way were included into the model. Fibroblast data were processed separately from the remaining data to contrast organismal with cell culture ageing. Normality of residuals was tested for each process using the Shapiro–Wilk test as implemented in R. Homogeneity of variance was tested using the Levene test from the 'car' package. If any of the two tests yielded $p \leq 0.005$ in the joint analysis across data sets, the respective process was not further considered. We used a cutoff of 0.005 rather than 0.05 to reduce the rate of false negatives. Nevertheless, our results did not change qualitatively, when a $p$-value cutoff of 0.05 was applied. For Gene Ontology 900 out of 1563, for KEGG Pathways 74 out of 135 and for Human Metabolic Pathways 43 out of 66 processes passed the model assumption tests. For model reduction in ANOVA, we used backward selection with a $p$-value threshold of 0.05. Interactions were iteratively reduced by starting with the highest order of interaction. Main effects or lower order interactions were only removed from the ANOVA model, if no higher order interaction in the model included this interaction or main effect. Model reduction continued until no influence with $p > 0.05$ remained or until the main effect 'age group' was removed. Reported FDR-corrected $p$-values correspond to the $p$-value of the main effect 'age group'. The final $p$-values for the main effect 'age group' are provided in Supplementary Data 2 for each ontology and each process. Supplementary Data 1 contains processes with an FDR-adjusted $p \leq 0.1$. The foldchange of a process was determined by subtracting the mean of the rank-normalized process activities of young samples from the mean of old samples (both old age groups combined). We verified that the ageing-regulation of identified processes was not influenced by obtaining samples from several tissues of the same animal (Supplementary Note 8). ANOVA was also performed as described above, but stratified for tissue (factors 'age group' and 'species') and stratified for species (factors 'age group' and 'tissue'). Processes were classified as part of the functional signature of ageing, if the following two conditions were met: First, a process needed to be significantly differentially regulated in the analysis of the unstratified data of all species combined. Second, the direction of a significantly differentially regulated process had to be consistent in at least one of the fish and one of the mammalian species. We confirmed that our approach only returned ageing-regulated processes by repeating the entire procedure 100 times with randomly reassigned age labels where we observed only few cases of significantly ageing-regulated processes (Supplementary Note 3). Moreover, we tested whether the similarity in ageing-regulated processes between species could be explained by a random overlap in significantly ageing-regulated processes between species. We found that the similarity of ageing-associated process regulation is unlikely to have arisen by chance alone thereby confirming a strong conservation of the functional signature of ageing (Supplementary Note 3).

We performed several sensitivity tests to assess the accuracy of the identified ageing regulated processes. Thus, we repeated the analysis using an approach for deriving process activity in which all genes were equally weighted. We determined the frequency of significantly differentially regulated genes in differentially regulated processes from the process-based analysis and assessed the effect of inclusion of all age groups in the analysis. We found that differentially regulated processes showed a high degree of similarity across these analyses (Supplementary Note 8).

**Comparison of ageing and disease signatures**. Gene expression data for ageing-associated diseases was obtained from various sources (Supplementary Note 2). Only data with age-matched cases and controls was used unless indicated otherwise (skin cancer, brain cancer, and leukemia). We determined differentially expressed genes in our ageing data between the two old age groups and the young age group as described in Supplementary Note 2. Previously reported differentially expressed genes during ageing were obtained from the original works (Supplementary Note 2). AMDA scores for a disease and an ageing data set were determined based on the comparison of normalized foldchanges between cases and controls in the disease data set between ageing-induced and repressed genes. Theoretically, the AMDA score can attain values between −2 and +2, while we only observed a maximal range between −0.34 and +0.52. For each AMDA score we determined the significance of ageing-mediated disease alignment using a Wilcoxon rank-sum test (Supplementary Note 8) and only considered them as significant if the FDR-corrected $p$-value of the test was smaller than 0.05. For more information, see Supplementary Note 8. We repeated the procedure 100 times after randomly swapping ageing-induced and ageing-repressed genes in the ageing data sets. We did not observe a single instance of significant ageing-mediated disease alignment in the randomized ageing data sets (Supplementary Note 8).

**Determination of disease alignment contribution scores**. Crude disease alignment contribution (DAC) scores between a process and a disease category were obtained by determining the shift in ageing-mediated disease alignment if genes of that process were removed. After normalization of crude DAC scores (division by number of comparisons and scaling to an absolute maximum of 1), they have a theoretical range of −1 to +1, with −1 corresponding to the process that most strongly contributes to the ageing-associated reversion of a disease signature and +1 corresponding to the process with strongest disease alignment. For more details, see Supplementary Note 8.

**R package for multi-species analysis**. We implemented our approach of a process-based analysis of gene expression data in the R package PRODEX (PROcess-based Differential EXpression analysis, see code availability). This package allows users to create the mapping of gene expression data onto process activity information for Gene Ontology processes and KEGG Pathway. In addition, PRODEX provides the basic functionality required for analyzing the generated process activity data.

**Genetic analyses**. Reported risk alleles from published genome-wide association studies were obtained from GWAS catalog[57] (release e87 dated 4 January 2017) containing a list of manually curated SNPs reported in genome-wide association studies that meet a specific list of eligibility criteria (see https://www.ebi.ac.uk/gwas/docs/methods). Risk alleles for different types of ageing-associated diseases were obtained through matching reported disease traits with a list of keywords. On the basis of this data, we determined sets of SNPs predisposing to the considered disease categories: cancer, cardiovascular diseases, neurodegenerative diseases and type 2 diabetes, the latter three also collectively considered as degenerative ageing diseases. Through pairwise comparison of SNP sets, we identified shared risk SNPs. Shared risk SNPs correspond either to identical risk SNPs between the two SNP sets or two SNPs that are co-inherited through strong linkage disequilibrium based on data from the 1000 genomes project[9]. On the basis of this information we identified synergistic risk SNPs as those where the same allele was a risk allele in both SNP sets and antagonistic risk SNPs as those where one allele at a genomic position was a risk allele in the first SNP set (i.e., a risk SNP for the first disease) and the alternative allele a risk allele in the other SNP set (i.e., a risk SNP for the other disease). For more information, see Supplementary Note 8. Please note that by using the GWAS catalog data, we combined data from a wide variety of human cohorts (Supplementary Data 1). This might bias results compared to the utilization of one single cohort with available genetic data covering all of the considered diseases with sufficient case numbers at once. However, such a data set is not available yet.

**Code availability**. Source code to reproduce the analysis reported in this paper as well as the associated data is available at datadryad.org (doi:10.5061/dryad.4b5n5).

**Data availability**. All RNA-seq data generated in this study were uploaded to NCBI's Gene Expression Omnibus: *Danio rerio* (GSE74244), fibroblasts (GSE63577 / GSE60883), *Homo sapiens* (GSE75337, GSE103232), *Mus musculus* (GSE75192, GSE78130), and *Nothobranchius furzeri* (GSE52462/GSE66712). Further details surrounding the accession numbers of the individual data sets are provided in Supplementary Data 1, sheet "Sample Overview".

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

## Acknowledgements

We thank Christina Ebert and Sabrina Stötzer for zebrafish and Biancka Lanich for *N. furzeri* maintenance. Moreover, we thank Sabine Gallert, Ivonne Görlich, Madlen Günther, Christin Hahn, Ivonne Heinze, Cornelia Luge and Sabine Matz for technical assistance. We thank Charles Harding for useful discussion, for providing us with the corrected survival data of the ED01 mouse study and a script for determining malignant transformation rates for this data. We are grateful to Friederike Flachsbart for helpful discussions and Allon Wagner, Noam Auslander and Sridhar Hannenhalli for critical reading of the manuscript and feedback. We acknowledge the Surveillance, Epidemiology, and End Results (SEER) Program (www.seer.cancer.gov), for providing data on cancer epidemiology, the Center for Disease Control and Prevention (www.cdc.gov) for providing information on causes of mortality as well as Tianxiao Huan and Daniel Levy for providing information on correlation of gene expression with hypertension (work funded by the National Institutes of Health / National Heart, Lung, and Blood Institute). We acknowledge funding by the German Ministry for Education and Research within the framework of the GerontoSys initiative (research core JenAge, funding code BMBF 0315581) to S.D., C.E., P.H., R.G., C.K., M.P., M.R., St.S., and J.S. as well O.W.W. and within the e:Med—GlioPATH project to Sa.S. (funding code 01ZX1402C). We thank the excellence cluster "Inflammation at Interfaces" (DFG support code EXC306) for support to C.K. Moreover, we acknowledge support by DFG FOR 1738 B2; DFG BU 1327/4-1; BMBF Bernstein Fokus (FKZ 01GQ0923); EU BrainAge (FP 7/HEALTH.2011.2.2.2-2 GA No.:279281); BMBF Irestra (FKZ 16SV7209), and the Swiss National Science Foundation (SNF 31003A_156031).

## Author contributions

Designed research: C.K., E.R., Sa.S., J.S., M.P., A.l.C., Am.C., S.D., C.E., C.F., P.H., M.K., M.R., St.S., R.G., and O.W.W. Conducted experiments: M.G., C.F., M.B., N.H., S.M., Si.S., J.M. Analyzed data: P.A.I., Sa.S., C.K., M.G., S.P., D.E., U.M., V.A., and R.K. Wrote paper: C.K., E.R., Sa.S., P.A.I., M.G., Am.C., and D.E. Contribution of model organism data: *D. rerio*: N.H. and C.E.; *N. furzeri*: M.B. and A.l.C.; *M. musculus* and *H. sapiens*: C.F., Si.S., J.M., and O.W.W.; *H. sapiens* fibroblasts: S.M., S.D., and P.H. RNA-seq data generation and processing: M.G., M.P., S.P., U.M., and R.G.

## Additional information

**Competing interests:** The authors declare no competing financial interests.

