## [Peer Review File · Nature Communications]

Reviewers' comments:

Reviewer #1 (Remarks to the Author):

In this manuscript, the authors focus on identifying transcriptional signatures associated with aging. Unlike existing studies that group aging diseases together the identify transcriptional signatures, the authors propose to group various aging disorders -- malignant and non-malignant diseases. Through this strategy the authors provide an interesting characterization of transcriptional signatures that differ between these diseases.

Major comments:

(1) It is not fully clear why the authors generate data from five time points but intentionally select just to use for their signatures.

(2) More data should be provided on the number of orthologs across all four species utilized and which biological processes lack consistent orthologs and therefore may not be fully represented in the analysis.

(3) The authors indicate that only two genes have been identified to date as associating with human longevity. This manuscript would be strengthened by demonstrating that they have identified additional genes (from their lists) that can be validated as associated with human longevity, maybe even at a greater level than the previous two genes.

(4) It would strengthen the manuscript to provide some experimental evidence demonstrating the antagonistic relationship of risk alleles with malignant and non-malignant diseases.

(5) It would be beneficial to the readers if the authors could address in the discussion any bias in biological processes that may be under-represented in existing ontologies (but relevant to aging, malignant disease, non-malignant diseases) and therefore bias the subsequent results.

Reviewer #2 (Remarks to the Author):

In the current manuscript the authors were seeking possible connections between alterations in aging-related disease epidemiology, and transitions in gene expression signatures and genomic variants of aging-related diseases and during "normal" aging, comparing cancer to non-cancer aging-associated conditions.

For this the authors generated a massive dataset comprised of aging-associated gene expression profiles and database from four species at different ages and four tissues. Additionally, to connect the gene expression profiles and corresponding biological processes to certain chronic diseases, the authors analyzed publicly available datasets originated from patients with most frequent aging-associated diseases.

The paper's main messages are:

- Authors generated a very detailed, standardized aging-related dataset from multiple organisms, tissues and ages. They identified aging-related gene expression signatures and biological processes which are conserved on a functional level both across species (including humans) as well as across tissues.
- The authors investigated the similarities between the gene expression signatures of aging- and disease-associated transcriptomic data sets. They found that while the "middle-aged" organisms show gene expression signatures reminiscent of cancer, the very old individuals have a gene expression profile similar to a "non-malignant" disease profile. The authors state that such a trade-off of gene

expression profile between “malignant” and “non-malignant” state is highly conserved.

- The authors investigated the correlations between (reported and newly identified) disease-associated genetic determinants (SNPs and gene variants) and the investigated “malignant and non-malignant” disease profiles. They found that certain genetic variants are antagonistically associated with malignant and non-malignant diseases.

General remarks:

The authors took tremendous effort to investigate the correlations between the aging epidemiology, aging-related diseases and normal aging on gene expression and genomic levels. They gathered an impressive amount of data from different sources to be able to arrive at general conclusions. The generated standardized data reinforced the connections between normal aging and the reported aging-related diseases at different life-stages on the gene expression level.

The generated data will be a useful resource for scientists across the aging field, and the standardized data generation together with the statistical methods used may be a guideline for similar large-scale projects in the future.

However, one limitation intrinsic to this type of study is that it is mainly inventory in nature and that the main conclusions are largely speculative as they are based on correlations, which are not experimentally addressed. The authors have not investigated the potential causes, regulators and biological mechanisms, which might be a driving force of the transition between the two main aging stages (i.e. upstream regulators, motif analysis, etc.). In addition, some interpretations of the findings are suboptimal as specified below in the specific comments.

Specific comments:

1. The authors use in their entire manuscript including the title the distinction between “malignant” and “non-malignant” states. This terminology is not very optimal. A “malignant” state suggests that the entire organism is in a malignant condition, whereas it takes only one cell to undergo critical mutations to initiate the process of carcinogenesis. So, in this view one can never deduce a malignant state from an expression profile of an entire organism or organ, which is comprised entirely or (almost) only of non-malignant cells. In contrast, for the non-cancer aging-associated diseases it takes a significant proportion of all cells to contribute to the disease state. For instance, recently a series of prominent papers has demonstrated that senescent cells, which accumulate with aging, significantly contribute to pathology, as apparent when they are selectively eliminated. It would be better to use other designations for the distinct aging states. For instance, in line with the down-regulation of cell cycle/replication-associated processes and the upregulation of several signaling pathways with aging it would be more appropriate to name them “proliferation-supportive” or “cell renewal” state versus “proliferation-inhibitory” or “cell preservative” states. The former state will also be supportive of cancer as this disease is stimulated by growth-promoting signals, whereas the latter state is consistent with the down-regulation of cell cycle parameters, upregulation of suppression of apoptosis and down regulation of the P53 class mediator (see Figure 2). Alternatively, one could call the first state a “cancer-permissive” phase in view of its pro-growth nature and the second phase a “chronic disease permissive” state (as cancer is in principle not a chronic disease), as apparent from the phenotype although it cannot be deduced from the expression and genomic data.

2. The cancer-associated gene expression signatures are dominated by proliferation-related pathways, which are also a general (i.e. not cancer linked) signature, that declines with age. This explains why oncogenes, often inducers of proliferation are in discordance and tumor suppressor genes, which often suppress growth, in concordance with aging-related signatures (lines 155-159). This is also consistent with the antagonistic relationship between risk SNPs for cancer as a proliferation disorder and neurodegenerative diseases, which concern non-dividing cells in figure 5A.

3. To which extent is cellular senescence as an important aging-associated process represented in the profiles? Senescence is not only an anti-tumor mechanism in contrast to what is stated in line 211. In fact, it has been demonstrated by Campisi and others that the microenvironment of senescent cells may promote cancer. In the aging-associated transcriptional changes in Figure 2 a large number of inflammatory pathways appear upregulated with aging, in agreement with the senescence-associated

secretory phenotype (SASP) which contains a strong cytokine component. Moreover, senescent cells are arrested in cell proliferation and suppress apoptosis, which is consistent with the pathways identified in this study to correlate with aging as a chronic-disease permissive state. This topic should be more extensively considered in the text.

4. One may a priori raise some questions with respect to the comparison done in S3.2 regarding analysis of somatic copy number variations derived from tumors in aging-induced and aging-suppressed genes. As tumors originate from individual cells which undergo a specific evolution based on genome instability, this is very different from non-cancer aging diseases, which are based on a significant proportion of cells from a given tissue.

5. The contribution of the *Nothobranchius furzeri* dataset to the total should be considered with some cautionary considerations. The biology and pathology of aging in this interesting but also unusual organism is less worked out compared to the other three main species and the type of aging-related processes may as a consequence at least in part be distinct from that of the other species. This aspect should be briefly mentioned in the paper.

6. The authors mainly focus on biological processes (determined by 5 or more genes per GO or KEGG category), it would be also interesting and important to see the role and contribution of individual genes in the mentioned pathways. It is not clear whether the same genes are re-occurring in a given pathway at the different time points and tissues /organisms. E.g. Spindler has identified a significant set of genes which are overrepresented in anti-aging treatments which might be of interest to analyze. Moreover, one expects that a few genes of a pathway determine the output and would be expected to be overrepresented in expression analysis.

Minor remarks:

1. Line 94: remove "... for the first time ...".

2. In the first paragraph of the "Results" mention not only the four species but also the 4 tissues used in this study.

In conclusion: this manuscript describes a very detailed, extensive and systematic study which is by its nature somewhat descriptive. It is based on a massive bioinformatics analysis of different stages of the medically important process of aging in four vertebrates and four tissues. The results are a valuable resource and may be a starting point for biological verification and mechanism-focused research. As specified above several main interpretations are questionable and will need revision.

Reviewer #3 (Remarks to the Author):

The authors have undertaken a study on age-expression signatures, and disease correlates thereof, that includes new multi-tissue RNA-seq data in several species, as well as an integration of a number of age-expression, expression-disease, and SNP-disease datasets from the public domain. The findings from the standpoint of ageing trade-offs between malignant and non-malignant disease are not exactly novel, having been described by others in population science and epidemiology settings (i.e., all of Figure 1 has essentially been produced several times by others using a variety of data sources). What is most novel here is the assembly of expression-aging signatures and their characterization with respect to model organisms, diseases of ageing and lifespan, and the suggestion of antagonism between malignant and non-malignant disease alleles (though this is also previously proposed). I have some major and minor concerns with the current manuscript:

1) One area of concern is either inadequate or possibly incomplete or inaccurate reference to a few important prior studies, and over-statement of the current study. Several statements are made: ... "To address this hypothesis, we generated the currently most comprehensive transcriptomic data set of ageing covering four tissues in four different vertebrate model organisms." ... "Previous studies have reported only little overlap in ageing-associated differential expression of individual genes between species 6,7." ... ""previous lack of a clear association between ageing-associated transcriptomic

changes with gene expression signatures of ageing diseases.” ... “These results provide, for the first time, a clear link...”. In relation to these statements I suggest the authors should generally tone down the strength of such statements and avoid over-generalizing relative to the prior work. Some specific suggestions follow:

a. I don’t think it is important to call this the most comprehensive transcriptomic data set of aging. There are more comprehensive datasets in other domains (e.g., in brain aging) or better-powered datasets in terms of large sample sizes, and other multi-tissue datasets (e.g., Glass et al skin, blood, LCLs in humans). Rather I think the focus here should be highlighting that moderately large multi-tissue datasets in multi-species was acquired with similar current technologies (i.e., RNA-seq and processing pipelines), and the approach to comparing to SNP and RNA disease data.

b. Fushan et al 2015 Aging Cell (PMID 25677554) is an important reference that is completely missed. In some ways, it is a more comprehensive dataset though the focus there is on small samples sizes across 33 species and the relationship to longevity

c. Van den Akker et al. 2014 Aging Cell (PMID 24119000) is another relevant study that used multiple datasets to construct aging signature models (employing PPI networks). While that work did not directly examine specific diseases they linked to survival at high age

d. Re: “a lack of prior clear association between transcriptomic changes and ageing diseases”, I think that is an incomplete characterization. There are prior specific examples of target genes that could be highlighted from candidate studies, but more broadly in large scale studies cited:

i. Peters et al. showed that transcriptomic aging signatures were associated with many quantitative correlates/risk factors of complex diseases (BP, lipids, BMI). Peters et al. also showed correspondence in directionality with known rare ageing diseases. Orthologous genes from that study were recently shown to shorten or extend lifespan in *C. elegans* (Sutphin et al. 2017)

ii. It is mentioned that prior studies examined aging disorders “en bloc”. However, Yang et al. analyzed GWAS data and OMIM (for rarer disorders). See Yang et al. Figure 4 which directly provides results of specific diseases with aging signatures

2) The primary analyses identify ontologies and pathways in the ageing and/or disease overlap signals but largely do not identify specific key driver genes. The enriched pathways and ontologies also seem largely concordant with prior theories and datasets on aging. Are there novel insights into specific genes or pathways that the authors can highlight? Is there any potential translational aspect to the findings?

3) There is very poor alignment of the JenAge human blood results with other datasets (i.e., Figure 3 and S6 Fig 4), whereas other large human blood datasets do show correspondence (cross-cohort, SAFHS). This is likely due to very small samples in JenAge (n=7; n=9 in age subgroups) which are insufficient given variability in outbred human samples, and poor statistics in such a small sample. It might be wise to remove that data or acknowledge the likely limited utility of that data.

4) Zebrafish data seems the least consistent of any species. Skin signatures are concordant with others in the AMDA, whereas brain signatures appear discordant and liver signatures unrelated. Is there a potential biological or technical explanation for this?

5) The limitations of the current study are essentially not acknowledged. This is an important element for any study, even more so for one that relies on so much disparate data. A list follows of some

analyses should be conducted in this manner.

Minor

- 1) In Supplemental tables Longitudinal aging (N. furzeri and mice) it would be helpful to label the meaning of each GO entry
- 2) Suppl Fig 3 – mislabeled as 3C when there is only a 3B label
- 3) S7.4 – blood pressure was excluded but hypertension included? I would submit that these variants are largely correlated and dichotomous hypertension studies have largely identified BP quantitative trait variation in those better powered studies, with few true risk variants. If you were looking for large effect/rare hypertension variants you might need further filtering strategies to differentiate the true ones
- 4) S8.1 – Figure 5 – as presented I have a little trouble understanding the meaning of the background shading versus the boxplots. Can you just clarify – background shading is actual observed and boxplots and outliers represent the distributions after randomization?

Reviewer #4 (Remarks to the Author):

This is a very interesting approach to examining transcriptomic signature of aging, where there is differential prevalence in age-related diseases by age group.

Overall comments: the supplementary notes/figures are not in order they are described – a little difficult to follow.

Identification of differentially regulated processes across species:

When calculating activity of a process, what happens if you have a combination of upregulated and down regulated genes within a process? Would the downregulated gene “lower” the overall activity of the process?

Methods – identification of differentially regulated process, “From the five groups (young, mature1, mature2, old1, old2, cf. Supplementary Fig. S1)” do you mean Table S2.1. The next figure referenced should be just Fig2, not 2A. Can you describe clearly what you mean by “the three age groups were considered in the analysis”? Do you mean, these age groups were included in the factor “Age group” in the ANOVA?

Why not consider including all the age groups in the model as categorical variable rather than making pair-wise comparison. I understand you see the most differences between the youngest and the oldest groups but you are losing power.

Can you indicate how many processes were tested and how many failed the model assumption tests?

Comparison of aging and disease signature:

What are the age ranges of “old individuals” the data from where the AMDA is calculated? I am encouraged that the AMDA for cancer in middle age is positive. Is there an AMDA for the middle age group for the chronic diseases? I’d expect at the younger age group the score would be negative then

positive from middle age onward?

Longitudinal analysis of aging cohorts

How relevant would disease expression scores from human tissue samples be for non-humans particularly when the aging genes across species lack consistency as you discuss in the paper?

Shared genetic risk allele

This is an interesting idea and certainly could be part of the reason for the lack of association between disease SNP and longevity. Generally for shared risk alleles, were the effects sizes comparable for malignant and non-malignant SNPs? I do think that there are many other reasons including the relatively small effect sizes and/or variance explained by individual SNPs.

Reviewer #1

In this manuscript, the authors focus on identifying transcriptional signatures associated with aging. Unlike existing studies that group aging diseases together to identify transcriptional signatures, the authors propose to group various aging disorders -- malignant and non-malignant diseases. Through this strategy the authors provide an interesting characterization of transcriptional signatures that differ between these diseases.

Major comments:

#1.1

It is not fully clear why the authors generate data from five time points but intentionally select just to use for their signatures.

Please see also the related response #4.4 to reviewer 4. In our initial analysis, we considered all age groups but found the strongest changes between the first and the two last time points. Thus, we chose the comparison between the young and the two old age states as reference for ageing-associated changes. To address this comment we additionally compared the ageing response with changes between all pairs of age groups. We found a strong concordance in ageing-associated regulation between individual age groups and the overall ageing signature. Thus, across all comparisons between individual age groups (e.g. young vs. mature₁, young vs. mature₂, etc.) and across all ontologies, we found 492 instances of significant differentially regulated processes. Out of these cases, 324 (66%) showed the same direction of regulation like the ageing signature, while six cases (1.2%) showed a regulation opposing the ageing signature. The remaining processes were not detected as part of the ageing signature. Moreover, including all age-groups in our ANOVA-based procedure for detecting differentially regulated processes strongly reduced the number of processes that passed the model assumptions for ANOVA (and hence could be tested). More details are discussed in response to comment #4.4 by the fourth reviewer. Thus, considering the comparison of the first versus the two oldest age groups as reference for ageing both reflects ageing-associated changes between individual age groups and increases the number of processes that can be considered in the analysis. The corresponding analyses are now discussed in more detail in the results and supplement (l. 130-132 and Supplementary Note S8.2 and Supplementary Figure S4).

#1.2 More data should be provided on the number of orthologs across all four species utilized and which biological processes lack consistent orthologs and therefore may not be fully represented in the analysis.

We thank the reviewer for this comment. To address this point we have checked whether we have lost a considerable number of processes due to the requirement of each considered process to contain at least five genes with measurable expression across all datasets. Considering humans, we found that 53 gene ontology (GO) terms with at least 20 annotated genes (12 of them with more than 50 annotated genes) were not considered due to this requirement. This is a rather small number compared to the total number of 1563 considered GO terms and hence there seems to be no bias in process consideration due to our requirements for determining process activity. In the list of processes, we find a wide variety of different processes without a clear overrepresentation of specific functions. We address this point now in more detail in Supplementary Note S3.2 and data from this analysis is provided in Supplementary Data S1.

#1.3 The authors indicate that only two genes have been identified to date as associating with human longevity. This manuscript would be strengthened by demonstrating that they have identified additional genes (from their lists) that can be validated as associated with human longevity, maybe even at a greater level than the previous two genes.

*As outlined in our manuscript, we think that the antagonism between cancer and degenerative diseases is one of the main reasons, why only a small number of genetic variants are associated with human longevity. To find more associations we would need a completely different study design with genomic information for long-lived humans and normal-aged controls. Previous studies examining human longevity found only few associations, although they investigated cohorts of up to 100.000 cases and controls¹. This is clearly beyond our current study, which focusses on transcriptomic alterations during ageing and their association to ageing diseases. On the other hand, our longitudinal data, in principle, allows to infer genes whose expression changes are correlated with lifespan. However, for our longitudinal data of *N. furzeri*, this has already been investigated in detail in a prior work² along with experimental validation while for the mouse data the number of samples (n=8 per age group) is not sufficient to reliably determine lifespan-associated genes.*

#1.4 It would strengthen the manuscript to provide some experimental evidence demonstrating the antagonistic relationship of risk alleles with malignant and non-malignant diseases.

Please see also the related response #2.1 to the second reviewer. We agree that experimental evidence of the effects of some of the identified antagonistic risk SNPs (cancer versus degenerative diseases) would strengthen our manuscript. However, there are several limitations to such an analysis. First, like for all genome-wide association studies, the reported SNPs are not necessarily the causative SNPs, but rather only within linkage to the causative variation (i.e. the reported SNP is co-inherited with the causative SNP). Second, most SNPs reside in non-coding regions, which makes a functional interpretation of the effects of the SNP very challenging. Thus, almost all GWAS studies report only on associations without any further functional follow up. This is probably best exemplified by the association between cardiovascular diseases and the ANRIL locus, which is one of the best studied loci, due to its very strong association with cardiovascular disease. After its first report more than ten years ago³, further studies e.g. in Nature showed functional effects⁴, which were later refuted⁵. Until now the functional effects of the corresponding SNPs are still only partially understood⁶.

Following up on the reviewer's suggestion as best as we can, however, we screened the corresponding literature for more information about the functional effects of all the 40 shared risk SNPs that we identified. We were not able to identify a single work with a functional follow up conclusively demonstrating the mechanism underlying the association between a SNP and the corresponding disease phenotype. This example, along with the frequent lack of functional characterizations of reported SNPs in GWAS studies, demonstrates the difficulty in identifying the mechanisms underlying the disease association in further studies. We now state that functional follow ups will be an important component of future work in the discussion (l. 368 - 372).

#1.5 It would be beneficial to the readers if the authors could address in the discussion any bias in biological processes that may be under-represented in existing ontologies (but relevant to aging, malignant disease, non-malignant diseases) and therefore bias the subsequent results.

We thank the reviewer for this suggestion. See also our related response to comment #1.2. While there might be some bias in the process-based comparison of ageing, the disease comparison also includes differentially expressed genes for the individual species/tissues that were derived directly from the gene expression data. Hence, any bias in the ontologies has only a small influence on the subsequent disease analyses. We formulate this now more clearly in the results section (l. 164-166).

Reviewer #2

In the current manuscript the authors were seeking possible connections between alterations in aging-related disease epidemiology, and transitions in gene expression signatures and genomic variants of aging-related diseases and during "normal" aging, comparing cancer to non-cancer aging-associated conditions.

For this the authors generated a massive dataset comprised of aging-associated gene expression profiles and database from four species at different ages and four tissues. Additionally, to connect the gene expression profiles and corresponding biological processes to certain chronic diseases, the authors analyzed publicly available datasets originated from patients with most frequent aging-associated diseases.

The paper's main messages are:

- Authors generated a very detailed, standardized aging-related dataset from multiple organisms, tissues and ages. They identified aging-related gene expression signatures and biological processes which are conserved on a functional level both across species (including humans) as well as across tissues.
- The authors investigated the similarities between the gene expression signatures of aging- and disease-associated transcriptomic data sets. They found that while the "middle-aged" organisms show gene expression signatures reminiscent of cancer, the very old individuals have a gene expression profile similar to a "non-malignant" disease profile. The authors state that such a trade-off of gene expression profile between "malignant" and "non-malignant" state is highly conserved.
- The authors investigated the correlations between (reported and newly identified) disease-associated genetic determinants (SNPs and gene variants) and the investigated "malignant and non-malignant" disease profiles. They found that certain genetic variants are antagonistically associated with malignant and non-malignant diseases.

General remarks:

The authors took tremendous effort to investigate the correlations between the aging epidemiology, aging-related diseases and normal aging on gene expression and genomic levels. They gathered an impressive amount of data from different sources to be able to arrive at general conclusions. The generated standardized data reinforced the connections between normal aging and the reported aging-related diseases at different life-stages on the gene expression level.

The generated data will be a useful resource for scientists across the aging field, and the standardized data generation together with the statistical methods used may be a guideline for similar large-scale projects in the future.

#2.1 However, one limitation intrinsic to this type of study is that it is mainly inventory in nature and that the main conclusions are largely speculative as they are based on correlations, which are not experimentally addressed.

We think that besides demonstrating the existence of an antagonism between malignant and degenerative diseases as part of the ageing process, the important contribution of our study lies in its ability to provide a point of reference for future studies investigating ageing-associated transcriptomic changes as well as the molecular processes driving them. Thus, we provide, as the reviewer describes, both a comprehensive uniformly generated data set as well as a toolset for its analysis in the context of ageing diseases. In consequence, we were able to show that there is a common denominator of vertebrate ageing across species as well as tissues and that this common denominator is strongly linked to ageing-associated diseases. While our results provide a large number of avenues for further experimental validation, we feel that any experiment that could be performed based on our results would

certainly only be able to provide validation for a very specific aspect of our work. Hence, this would require us to bias our work toward a very specific aspect of ageing rather than the broad perspective that we are considering presently. Instead, based on our results, we are in the process of planning and executing experiments that follow up on the present study and are investigating how the antagonism between cancer and degenerative diseases manifests in the context of specific model systems of ageing diseases. We more specifically emphasize this point now in the discussion (l. 368-372).

#2.2 The authors have not investigated the potential causes, regulators and biological mechanisms, which might be a driving force of the transition between the two main aging stages (i.e. upstream regulators, motif analysis, etc.).

We thank the reviewer for raising this important point. Please note that the functional analysis based on DAC scores allowed us to identify a number of molecular processes that are key contributors to the antagonism between malignant and degenerative diseases. Moreover, our analysis on the genetic level points toward proteins such as p16Ink4a as well as Lnk as key mediators of the observed antagonism. To gain further insights into the drivers of the antagonism between cancer and degenerative diseases, we have performed, as suggested, a motif enrichment analysis of ageing regulated gene sets. Thereby, we have identified E2F1, NF-AT, AP1 and CEBPB as key regulators of the corresponding genes. Intriguingly, these transcription factors are known to play key roles in the pathogenesis of cancer as well as degenerative ageing diseases and they are centrally involved in the regulation of immune as well as cell cycle processes. These proteins hence also represent important targets of follow up studies investigating the specific molecular mechanisms that mediate the antagonism between cancer and degenerative diseases during ageing. We now discuss these results in more detail in context of the functional characterization of the ageing response (l. 245-251).

In addition, some interpretations of the findings are suboptimal as specified below in the specific comments.

Specific comments:

#2.3 The authors use in their entire manuscript including the title the distinction between "malignant" and "non-malignant" states. This terminology is not very optimal. A "malignant" state suggests that the entire organism is in a malignant condition, whereas it takes only one cell to undergo critical mutations to initiate the process of carcinogenesis. So, in this view one can never deduce a malignant state from an expression profile of an entire organism or organ, which is comprised entirely or (almost) only of non-malignant cells. In contrast, for the non-cancer aging-associated diseases it takes a significant proportion of all cells to contribute to the disease state. For instance, recently a series of prominent papers has demonstrated that senescent cells, which accumulate with aging, significantly contribute to pathology, as apparent when they are selectively eliminated. It would be better to use other designations for the distinct aging states. For instance, in line with the down-regulation of cell cycle/replication-associated processes and the upregulation of several signaling pathways with aging it would be more appropriate to name them "proliferation-supportive" or "cell renewal" state versus "proliferation-inhibitory" or "cell preservative" states. The former state will also be supportive of cancer as this disease is stimulated by growth-promoting signals, whereas the latter state is consistent with the down-regulation of cell cycle parameters, upregulation of suppression of apoptosis and down regulation of the P53 class mediator (see Figure 2). Alternatively, one could call the first state a "cancer-permissive" phase in view of its pro-growth nature and the second phase a "chronic disease permissive" state (as cancer

is in principle not a chronic disease), as apparent from the phenotype although it cannot be deduced from the expression and genomic data.

We thank the reviewer for this valid criticism. Please see also our related response to comments #2.4 and #2.5. We do not want to imply that an organism can be in a malignant or non-malignant state and thus we have tried to put emphasizes (already in the originally submitted manuscript) to state that the gene expression signature of ageing does not shift into but toward a disease signature. To make this distinction clearer, we now specifically state that we analyzed shifts toward but not into disease signatures (l. 158-160, l. 173). Moreover, we replaced the term “non-malignant diseases” with “degenerative diseases” throughout the manuscript to avoid the impression that an organism can be in either a malignant or non-malignant state. We feel that the alternative terms suggested by the reviewer cover only specific aspects of the disease antagonism that we report as the identified signature of ageing is not solely limited to proliferation-associated pathways (see also the response to comment 2.4). Moreover, “cancer permissive” or “chronic disease permissive” implies causation, or enabling, which we want to avoid as we can only report on associations, as the reviewer writes, but no causal relationships.

#2.4 The cancer-associated gene expression signatures are dominated by proliferation-related pathways, which are also a general (i.e. not cancer linked) signature, that declines with age. This explains why oncogenes, often inducers of proliferation are in discordance and tumor suppressor genes, which often suppress growth, in concordance with aging-related signatures (lines 155-159). This is also consistent with the antagonistic relationship between risk SNPs for cancer as a proliferation disorder and neurodegenerative diseases, which concern non-dividing cells in figure 5A.

We thank the reviewer for this very good suggestion. Indeed, as the reviewer points out, a decrease in cellular proliferation as a general consequence of ageing might be the underlying cause of the anti-malignant signature that we observe. To address this point, we recomputed AMDA scores without genes that are significantly differentially expressed during cellular senescence (based on our expression data from senescent cell cultures) and without all genes belonging to GO terms containing the term “proliferation”. After excluding these genes from the analysis, we still observed a reversion of cancer-associated gene expression and an alignment with expression changes of degenerative ageing diseases by the ageing signature. This supports that while cellular senescence and proliferation-associated processes are certainly important contributors to the observed antagonism, there are also further contributing processes. The corresponding results are provided in Supplementary Figure S4 and are discussed in the main manuscript (l. 178-185).

#2.5

3. To which extent is cellular senescence as an important aging-associated process represented in the profiles? Senescence is not only an anti-tumor mechanism in contrast to what is stated in line 211. In fact, it has been demonstrated by Campisi and others that the microenvironment of senescent cells may promote cancer. In the aging-associated transcriptional changes in Figure 2 a large number of inflammatory pathways appear upregulated with aging, in agreement with the senescence-associated secretory phenotype (SASP) which contains a strong cytokine component. Moreover, senescent cells are arrested in cell proliferation and suppress apoptosis, which is consistent with the pathways identified in this study to correlate with aging as a chronic-disease permissive state. This topic should be more extensively considered in the text.

The ageing signature that we have observed contains indeed many components associated with cellular senescence such as suppression of cellular proliferation and induction of immune-associated processes (as part of the senescence-associated secretory phenotype). However, as pointed out in the response to comment #2.4, we still observed a reversion of cancer-associated gene expression changes and an alignment with degenerative ageing diseases after removing genes associated with cellular senescence. Moreover, in the comparison of process regulation between the different datasets included in our study, senescent cell cultures showed the strongest discordance with the general ageing signature (c.f. Figure 2, column “H.s. fibroblasts” and Supplementary Figure S3). We now address the relationship between the identified expression signature of ageing and cellular senescence in more detail in the main manuscript (l. 178-185 and Supplementary Figure S4). Moreover, we mention that senescent cells can promote the transition of pre-malignant cells to full malignancy (l. 242-244).

#2.6

One may a priori raise some questions with respect to the comparison done in S3.2 regarding analysis of somatic copy number variations derived from tumors in aging-induced and aging-suppressed genes. As tumors originate from individual cells which undergo a specific evolution based on genome instability, this is very different from non-cancer aging diseases, which are based on a significant proportion of cells from a given tissue.

In this analysis, we showed that ageing induced genes were enriched with genes that are frequently lost during tumorigenesis. This is an additional approach to identify potential tumor suppressors that are lost during cancer genome evolution. Thus, we did not relate these observations to degenerative ageing diseases.

#2.7

The contribution of the *Nothobranchius furzeri* dataset to the total should be considered with some cautionary considerations. The biology and pathology of aging in this interesting but also unusual organism is less worked out compared to the other three main species and the type of aging-related processes may as a consequence at least in part be distinct from that of the other species. This aspect should be briefly mentioned in the paper.

*N. furzeri is indeed an emerging model organism and the number of aging phenotypes investigated is relatively limited in comparison to well-established models. Yet, all aging-related phenotypes investigated up to date at the behavioral, histopathological and molecular/cellular levels are consistent with a typical vertebrate pattern (reviewed in Cellerino et al., 2016⁷). Moreover, in our analysis we found that *N. furzeri* shows a very similar molecular signature of ageing compared to the other vertebrates, on the level of differentially regulated processes with ageing as well as the association of these changes with ageing diseases. We now mention this point more explicitly (l. 120).*

#2.8

6. The authors mainly focus on biological processes (determined by 5 or more genes per GO or KEGG category), it would be also interesting and important to see the role and contribution of individual genes in the mentioned pathways. It is not clear whether the same genes are re-occurring in a given pathway at the different time points and tissues /organisms. E.g. Spindler has identified a significant set of genes which are overrepresented in anti-aging treatments which might be of interest to analyze. Moreover, one expects that a few genes of a pathway determine the output and would be expected to be overrepresented in expression analysis.

We thank the reviewer for bringing up this point. Please see also the response to comments #4.2 and #4.7 of the fourth reviewer. To address this comment, we tested which genes belonging to differentially expressed processes were differentially expressed with age. We found that for 138 out of 171 processes, genes belonging to that process were preferentially differentially regulated in the direction of change of process activity. Thus, changes in the processes were not necessarily dominated by few highly abundant genes but the majority of genes changed expression in the direction of the process. These results are now discussed in more detail in the methods section (l. 520-525) and Supplementary Material S8.2 and results of the analysis provided in Supplementary Data S1.

With respect to the question whether related genes are differentially expressed across species and tissues, we now also provide a list of genes whose orthologues are frequently differentially regulated across all species and tissues in Supplementary Data S1 (the gene set which we used for motif enrichment analysis). To this end, differentially expressed genes in each species and tissue from the JenAge data (13 data sets) were mapped to their corresponding human orthologues. Then we counted the number of cases of significant up- and down-regulation for each gene and used a cut-off of at least five more cases of significant up- versus down-regulation to determine genes with conserved regulation. Thereby, we identified a list of 896 genes with conserved regulation according to our criteria (Supplementary Data S1). By performing gene set enrichment analysis on this gene set, we found that down-regulated genes were highly enriched for cell cycle-associated processes while up-regulated genes were enriched for immune associated processes. However, since these results are concordant with the results we obtain from testing ageing-associated changes in process activity directly, we have not included them in the manuscript due to redundancy. We now discuss the related analyses in more detail in Supplementary Note S2.2.

Minor remarks:

1. Line 94: remove "... for the first time ...".

The statement was removed.

2. In the first paragraph of the "Results" mention not only the four species but also the 4 tissues used in this study.

This information was added (l. 113).

In conclusion: this manuscript describes a very detailed, extensive and systematic study which is by its nature somewhat descriptive. It is based on a massive bioinformatics analysis of different stages of the medically important process of aging in four vertebrates and four tissues. The results are a valuable resource and may be a starting point for biological verification and mechanism-focused research. As specified above several main interpretations are questionable and will need revision.

Reviewer #3

#3.1

The authors have undertaken a study on age-expression signatures, and disease correlates thereof, that includes new multi-tissue RNA-seq data in several species, as well as an integration of a number of age-expression, expression-disease, and SNP-disease datasets from the public domain. The findings from the standpoint of ageing trade-offs between malignant and non-malignant disease are not exactly novel, having been described by others in population science and epidemiology settings (i.e., all of Figure 1 has essentially been produced several times by others using a variety of data sources). What is most novel here is the assembly of expression-aging signatures and their characterization with respect to model organisms, diseases of ageing and lifespan, and the suggestion of antagonism between malignant and non-malignant disease alleles (though this is also previously proposed). I have some major and minor concerns with the current manuscript:

Indeed some of our observations on the epidemiology of ageing diseases have already been discussed in previous works, especially with respect to the decline in cancer incidence with old age. However, many works on ageing diseases implicitly or explicitly assume that the incidence of all ageing diseases including cancer increases exponentially with age (see e.g. Berger et al. [2006]⁸ or Magalhaes [2013]⁹). Thus, we think it is important not only to reference previous works showing that cancer incidence does not exponentially increase with age but explicitly show the difference between cancer and degenerative disease epidemiology using actual epidemiological data. Furthermore and importantly, our study shows that the tradeoff observed on the epidemiological level is accompanied and associated with a parallel inverse relationship occurring at the expression, molecular level. We reformulated the corresponding sections in the introduction to make this point clearer (e.g. l. 65, l. 81).

#3.2

1) One area of concern is either inadequate or possibly incomplete or inaccurate reference to a few important prior studies, and over-statement of the current study. Several statements are made: "To address this hypothesis, we generated the currently most comprehensive transcriptomic data set of ageing covering four tissues in four different vertebrate model organisms." ... "Previous studies have reported only little overlap in ageing-associated differential expression of individual genes between species 6,7." ... ""previous lack of a clear association between ageing-associated transcriptomic changes with gene expression signatures of ageing diseases." ... "These results provide, for the first time, a clear link...". In relation to these statements I suggest the authors should generally tone down the strength of such statements and avoid over-generalizing relative to the prior work. Some specific suggestions follow:

We thank the reviewer for this valid criticism and apologize for the inappropriate wording. We have reworded the corresponding sections as suggested.

#3.3

a. I don't think it is important to call this the most comprehensive transcriptomic data set of aging. There are more comprehensive datasets in other domains (e.g., in brain aging) or better-powered datasets in terms of large sample sizes, and other multi-tissue datasets (e.g., Glass et al skin, blood, LCLs in humans). Rather I think the focus here should be highlighting that moderately large multi-tissue datasets in multi-species was acquired with similar current

technologies (i.e., RNA-seq and processing pipelines), and the approach to comparing to SNP and RNA disease data.

We thank the reviewer for this suggestion. We have reworded the corresponding paragraphs accordingly (l. 116-117) and also referenced the indicated work (l. 53).

#3.4

b. Fushan et al 2015 Aging Cell (PMID 25677554) is an important reference that is completely missed. In some ways, it is a more comprehensive dataset though the focus there is on small sample sizes across 33 species and the relationship to longevity.

We thank the reviewer for raising this point. While we were aware of this work, we did not cite it in the previous manuscript, since it contains only a single time point of expression for each species and tissue and hence it is not possible to assess ageing-associated expression changes in the individual species and tissues. We however acknowledge the importance of this work and now cite it in the introduction as an example for the study of the relationship between gene expression and life-history traits across species (l. 50-51).

#3.5

c. Van den Akker et al. 2014 Aging Cell (PMID 24119000) is another relevant study that used multiple datasets to construct aging signature models (employing PPI networks). While that work did not directly examine specific diseases they linked to survival at high age.

We thank the reviewer for pointing us to this study, which we now reference in the introduction (l. 54).

#3.6:

d. Re: "a lack of prior clear association between transcriptomic changes and ageing diseases", I think that is an incomplete characterization. There are prior specific examples of target genes that could be highlighted from candidate studies, but more broadly in large scale studies cited:

We agree with the reviewer and have toned down this statement accordingly.

#3.7

i. Peters et al. showed that transcriptomic aging signatures were associated with many quantitative correlates/risk factors of complex diseases (BP, lipids, BMI). Peters et al. also showed correspondence in directionality with known rare ageing diseases. Orthologous genes from that study were recently shown to shorten or extend lifespan in *C. elegans* (Sutphin et al. 2017)

We have now added these references (l. 54).

#3.8

ii. It is mentioned that prior studies examined aging disorders "en bloc". However, Yang et al. analyzed GWAS data and OMIM (for rarer disorders). See Yang et al. Figure 4 which directly provides results of specific diseases with aging signatures

While Yang et al analyzed the overlap between genes that are differentially expressed with age with genes that are associated with human diseases, this prior analysis has several shortcomings:

- 1) As *Yang et al.* point out themselves, some of the reported associations might represent false positives, as they did not correct for multiple testing. Indeed, when performing multiple test corrections on the p-values they report (using false discovery rate control), only 19 of the reported 210 significant associations between ageing-regulated gene sets and diseases remain.
- 2) In contrast to our analysis, the analysis of *Yang et al.* is not able to account for the direction of effects with respect to ageing-associated gene expression changes, since they only test for the enrichment of disease-associated genes in ageing-regulated genes. Thus, they are not able to infer in which sense ageing-associated transcriptional changes are related to these diseases.

From their analyses, *Yang et al.* conclude that “Overall our results suggest that the connections between aging and diseases are very complex. Although we observed some direct connections between disease and tissue type, many connections could be indirect and thus undetectable from simple enrichment analysis.” We cited their conclusion as one recent example that shows that the relationship between ageing diseases and ageing-associated transcriptomic changes is still poorly understood.

#3.9

2) The primary analyses identify ontologies and pathways in the ageing and/or disease overlap signals but largely do not identify specific key driver genes. The enriched pathways and ontologies also seem largely concordant with prior theories and datasets on aging. Are there novel insights into specific genes or pathways that the authors can highlight? Is there any potential translational aspect to the findings?

Please see also our response #2.1 and #2.2 to reviewer 2. We have now included an additional analysis that, beyond the genes that we identified in the genetic analysis, points to several transcription factors that are key mediators of the antagonism between cancer as well as degenerative diseases and are centrally involved in the regulation of cell cycle and immune processes. As the reviewer indicated, almost all of the identified pathways have already been highlighted or discussed in the quite extensive existing literature. Thus, we think that it is more important to validate the ageing-regulation of the processes that we have identified (based on their identification in previous analyses), point to those that are actually part of the conserved core of ageing regulation, and emphasize those that change in opposing ways in the degenerative vs malignant disorders, which is the main focus and novelty of our work. We now point this out more clearly in the discussion (l. 368-372).

#3.10

3) There is very poor alignment of the JenAge human blood results with other datasets (i.e., Figure 3 and S6 Fig 4), whereas other large human blood datasets do show correspondence (cross-cohort, SAFHS). This is likely due to very small samples in JenAge (n=7; n=9 in age subgroups) which are insufficient given variability in outbred human samples, and poor statistics in such a small sample. It might be wise to remove that data or acknowledge the likely limited utility of that data.

Due to the valid criticism of the reviewer, we have included eight additional samples from human blood for each of the four age groups in humans which we recently sequenced (32 samples in total). Additionally, we recomputed all differentially expressed genes using *DESeq2*, instead of the rather outdated *edgeR* and *DESeq* packages that we used in our initial submission. Based on the new, extended samples (now encompassing 15 samples for blood per age group), we observed a considerable number of statistically significant differentially expressed genes in our blood data (102/293 genes significantly up-/down-

regulated instead of 36/110, c.f. Supplementary Note S2.1 and S2.2). With the extended blood data, we observed a significant alignment, as well as reversion of ageing disease expression signatures also based on our blood transcriptomic data (Fig. 3). All analysis in the manuscript have been updated based on this extended data set.

#3.11

4) Zebrafish data seems the least consistent of any species. Skin signatures are concordant with others in the AMDA, whereas brain signatures appear discordant and liver signatures unrelated. Is there a potential biological or technical explanation for this?

As outlined in the previous comment, we recomputed differentially expressed genes using DESeq2. Based on these gene sets, we observed an AMDA signature for D. rerio liver that aligns the transcriptome with degenerative diseases, but opposes cancer-associated gene expression signature. We can only speculate why we observe a discordant signature for D. rerio brain and hence do not discuss this point in more detail in the manuscript as it concerns only a single tissue in a single species.

5) The limitations of the current study are essentially not acknowledged. This is an important element for any study, even more so for one that relies on so much disparate data. A list follows of some important limitations to consider and either try to address, or at least acknowledge

We thank the reviewer for her/his clear criticism and apologize for lack of clarity. We have addressed the mentioned points as described below.

#3.12

a. Correlation (or cross-sectional association) does not imply causation. While there is some longitudinal data included (n=45 killfish – 2 ages; n=8 mouse ear punch – 2 ages) there is none for humans, and the case-control data and most of the other datasets from the public are cross-sectional. Thus, there are a variety of potential sources of confounding that are difficult to guard against and the findings may partially or largely represent ageing and disease responsive signatures rather than causal ones

We thank the reviewer for this comment. While we have included longitudinal data for fish and mouse, longitudinal data for humans is not available likely due the required timespan between sampling time points that would be necessary to obtain meaningful data. This longitudinal data and its similarity to the ageing response in cross-sectional data supports the assumption that the observed changes cannot solely be explained by cohort effects (i.e. differences in gene expression between age groups are only due to more susceptible individuals dying first). Indeed, most prior studies on the ageing transcriptome have exclusively looked into cross-sectional data without attempting to confirm the identified changes in longitudinal analyses. Moreover, we observed similar associations between ageing and diseases when excluding the oldest age group in which a considerable fraction of the population has already died (Supplementary Note S7). Finally, the observation that the antagonism between cancer and degenerative diseases is also visible on the genetic level and there is an association to similar processes (cell cycle and immune regulation) strongly argues against population heterogeneity as a dominant confounding factor. We now address these limitations including the fact that we report on correlation rather than causation in more detail in the manuscript (l. 201-205, l. 207-209, l. 212, l. 368-372).

#3.13

b. The approach of applying ANOVA based analyses across several age groups is somewhat limited (and might be better powered and outperformed by linear regression or non-linear models if adequate samples exist across the ageing distribution). One relevant discussion and work to consult implies non-linear models may be the best approach in age-expression research – see Gheorghe et al. BMC Genomics 2014

While linear regression or non-linear models might enable the detection of more differentially regulated processes and genes we think that our ANOVA-based procedure is more accurate here for two reasons:

- 1) We sampled data for precisely defined age groups. Thus, our samples are not equally distributed across the entire lifespan of the organisms but originate from “only” five time-points of ageing. This precludes application of the method described by Gheorge et al. since it requires an even distribution of samples across all ages.*
- 2) The ANOVA-based procedure can explicitly take into account the origin of data from different species and tissues. Hence, all data sets are considered at once while the analysis presented in the referenced paper only considers the different data sets individually.*

#3.14

c. Did the ageing-disease models account for competing risks or do they assume only single disease/single morbid sources? If not, the potential influences on accuracy and incidence estimates should be acknowledged

Since we used published information on risk SNPs, we have to rely on the original studies to account for competing disease risks. In principle, to derive precise estimates about the strength of interactions between diseases for specific SNPs and account for competing risks, we would need a cohort for which information on all of the considered diseases is available and there is a sufficiently large sample population to appropriately cover these diseases (which is, however, not available yet). Since we used published information from a diverse set of cohorts, this represents a potential source of confounding which we acknowledge in the manuscript (l. 578-581).

#3.15

6) Does Figure 2 include in the heatmap correlations with other datasets? Please clarify. If not, this would be of interest. It is written that "Close inspection of this functional ageing signature reveals a high concordance with previously reported ageing-associated transcriptional changes in individual species and tissues." However, it is not clear to me if this comparison was driven by a quantitative analysis or interpretation. [in the AMDA heatmaps things are clearly shown for the other datasets beyond JenAge]

We apologize for the ambiguous formulation. Figure 2 only depicts changes in process activity based on the JenAge data. To address the question raised by the reviewer, we performed a pairwise comparison of all ageing-regulated gene sets considered in the disease analysis (which also includes previously published gene sets). In general, we find a strong concordance in ageing-regulated genes sets across species and across tissues also encompassing previously reported ageing-regulated gene sets. The corresponding analysis is now described in Supplementary Material S3.1. Moreover, we clarified the corresponding statement in the main manuscript (l. 132, l. 141-142).

7) I do have some concerns over the disease datasets included in the AMDA and DAC as follows:

#3.16

a. The cancer samples come from various cellular origins, whereas the vast majority of other disease samples come from blood/white blood cells. Could this not bias toward less concordance of age-expression signatures with disease expression signatures as seems to be observed?

We thank the reviewer for pointing this out. Indeed, most of the cancer samples originated from tissue (cancer versus healthy tissue from the same donor), while samples for degenerative diseases, especially for neurodegenerative diseases (except one sample) and cardiovascular diseases, originated from blood (except two samples). For type 2 diabetes most samples originated from involved tissue (pancreatic islets, liver and adipose tissue). In the case of cardiovascular diseases, the utilization of blood samples likely does not represent a problem, due to the direct involvement of immune cells from the blood in the pathology of the disease. In general, while we observed the largest fraction of significant AMDA scores between cancer and the ageing data (256 out of 336 comparisons, 76%), only marginally smaller numbers were reached in the other diseases: 118 out of 168 (70%) for cardiovascular diseases, 74 out of 120 for neurodegenerative diseases (62%) and 68 out of 96 for type 2 diabetes (71%). Thus, in general we do not see a drastic decrease in significant associations between ageing and ageing diseases due to the consideration of blood-based data. We now explicitly mention the origin of the disease samples in the main manuscript (l. 168-169).

#3.17

b. Further, samples from cancer histology are less likely to be pure cell populations which may make their signals less ubiquitous

To investigate the effect of the tumor cell ratio, we compared differentially expressed genes between control and tumor using samples with different cellularity. We found that differentially expressed genes between tumor samples with different levels of cellularity were mostly concordant. Hence, we assume that cellularity has only a small influence on our results. Details about this analysis are now provided in Supplementary Material S2.3.

#3.18

c. I am somewhat surprised at the modest sample size in expression-disease datasets observed S2.3. How can sample sizes of 3 cervical cancers, 3 pancreatic cancers, 9 HF cases, 7 livers in diabetics be useful? There likely should be a minimum cutoff exercised. I am also surprised that for some common complex diseases (MI, Stroke, T2D, Alzheimer's disease) larger sample sizes could not be located? Is this because larger studies did not deposit results in repositories?

We thank the reviewer for raising this important point. We agree that just three samples might be too small to draw reliable conclusions about disease-specific expression changes. Thus, we removed ICGC data from cervical and pancreatic cancer from the analysis (both with only three cases and three controls). For pancreatic cancer we found an alternative study that included larger sample numbers (45 cases and 45 matched controls), but for cervical cancer we were not able to locate a study with more samples. In general, we

searched Gene Expression Omnibus with the respective disease term for suitable studies and considered studies with the highest sample numbers first. In the case of heart failure, this data set included longitudinal data for several time points after admission to hospital and we only used data for the first day of admission which explains the low number of cases despite this study containing the largest number of samples for heart failure in GEO. Thus, we checked additional studies related to heart failure on GEO and found a study of 177 cases and 136 controls that is now included in the analysis instead of the original data. For type 2 diabetes the low number of samples was specific to the liver samples as one affected tissue (for pancreatic islets, sample numbers are larger). We were not able to locate a study with larger sample sizes for liver in type 2 diabetes patients. For stroke, two larger studies exist on GEO, however one of them does not contain age-matched controls (GSE58294) and the other does not include information on the case/control status of the samples (GSE47728). For the other mentioned diseases, cohorts typically included at least 40 cases and controls and in most cases more than hundred samples. While larger cohorts with gene expression data might exist, the corresponding expression data is at least not deposited in GEO and we think that the cohort size should be sufficient to reflect disease-specific gene expression signatures. We have reperformed all downstream analysis with the updated disease data sets.

#3.19

8) In the Synergistic and Antagonistic GWAS analysis can you be confident that you have established modeling of the same allele in each study? That is, is there 100% confidence in the reported strand-specificity and modeled allele in each study included? If not, A/T and C/G alleles can easily be confused and conclusions about synergism vs. antagonism potentially confounded.

We used the strand information provided by GWAS catalog and thus assume that it is correct (see <https://www.ebi.ac.uk/gwas/docs/methods> for more information about the manual SNP curation procedure used by this database). We now outline this in more detail in the methods section (l. 565-566).

#3.20

9) In the GWAS comparison a concern is the authors used the SNP rather than independent signal as the unit of comparison/enrichment. In this way some loci will be counted multiple times whereas they really represent a single antagonistic/synergistic relationship. For one example, on chromosome 4 you have a single SNP associated with breast cancer and then 5 correlated SNPs associated. The appropriate unit of comparison is independent loci/signals after LD correlation reduction and those analyses should be conducted in this manner.

While an analysis on the level of independent genomic loci would be preferable, we think that a SNP-based analysis is more relevant here for several reasons:

- 1) A genomic locus might contain both antagonistic and synergistic risk SNPs as is the case, for instance, for locus 5 in the comparison of degenerative ageing diseases among each other. Thus, in such a scenario it is difficult to define whether a locus represents an antagonistic or a synergistic interaction.*
- 2) If several SNPs have been reported for a single locus, they originate from independent studies. Thus, if a genomic locus contains several shared risk SNPs these associations are much stronger (i.e. disease associations reported in several independent studies) compared to a locus that just contains a single reported SNP for a study. Performing the analysis just on the level of independent loci would, in our*

view, bias results since we would implicitly assume that each locus is equally strongly supported by previous data. For instance, the locus close to the senescence regulator p16Ink4a contains 12 shared antagonistic risk SNPs and is also the strongest known genomic risk locus for cardiovascular diseases. In contrast, two of the synergistic risk SNPs between cancer and degenerative diseases are the only shared associations within their respective locus. Please note, however, that we do not report the same risk SNP several times if it has been reported in independent studies.

To address this concern, we now first mention the number of independent loci that we have identified before discussing the antagonistic and shared risk SNPs that are contained within them (l. 291, l. 305-307). Moreover, we provide the number of independent loci in the figure comparing the number of synergistic and antagonistic risk SNPs (Fig. 5A). Information on the genomic loci to which each risk SNP belongs is provided in Supplementary Data S1.

Minor

#3.21

1) In Supplemental tables Longitudinal aging (N. furzeri and mice) it would be helpful to label the meaning of each GO entry

This information is now provided.

2) Suppl Fig 3 – mislabeled as 3C when there is only a 3B label

Corrected.

#3.22

3) S7.4 – blood pressure was excluded but hypertension included? I would submit that these variants are largely correlated and dichotomous hypertension studies have largely identified BP quantitative trait variation in those better powered studies, with few true risk variants. If you were looking for large effect/rare hypertension variants you might need further filtering strategies to differentiate the true ones

We thank the reviewer for pointing this out and apologize for the lack of clarity. In the GWAS analysis we just included binary and no quantitative traits. There are some GWAS analyses that consider hypertension as a binary trait (patient has hypertension/has no hypertension) while blood pressure is always reported as a quantitative trait. We mentioned blood pressure simply to exemplify this. To clarify this point, we removed blood pressure as an example for a quantitative trait and now only state that we included only binary traits.

#3.22

4) S8.1 – Figure 5 – as presented I have a little trouble understanding the meaning of the background shading versus the boxplots. Can you just clarify – background shading is actual observed and boxplots and outliers represent the distributions after randomization?

As the reviewer correctly assumes, background shading corresponds to actual cases of conserved and opposing regulation while the boxplots show the distribution after randomization. We reworded the description of the figure to make this point clearer.

Reviewer #4

This is a very interesting approach to examining transcriptomic signature of aging, where there is differential prevalence in age-related diseases by age group.

#4.1

Overall comments: the supplementary notes/figures are not in order they are described – a little difficult to follow.

We corrected the order of the Supplementary Notes and Figures.

#4.2

Identification of differentially regulated processes across species:

When calculating activity of a process, what happens if you have a combination of upregulated and down regulated genes within a process? Would the downregulated gene “lower” the overall activity of the process?

The down-regulated genes would actually lower the activity of a process as long as the sum of expression values across all genes belonging to a process is reduced. In general, if a process contains up- as well as down-regulated genes, our summation-based approach would infer a down-regulation as long as the sum of expression values across all genes is significantly reduced. To test whether up- or down-regulation of processes might be mostly driven by few highly abundant or a few unexpressed genes, we tested for each of the inferred ageing-regulated processes whether their constituting genes are also predominantly regulated in the direction of regulation of the process. For 138 of 171 ageing-regulated processes, significantly differentially regulated genes changed transcription preferentially in the same direction as the overall activity change of the process. We address this point in more detail in the methods section of the manuscript (l. 520-525) and Supplementary Note S8.2.

#4.3

Methods – identification of differentially regulated process, “From the five groups (young, mature1, mature2, old1, old2, cf. Supplementary Fig. S1)” do you mean Table S2.1. The next figure referenced should be just Fig2, not 2A. Can you describe clearly what you mean by “the three age groups were considered in the analysis”? Do you mean, these age groups were included in the factor “Age group” in the ANOVA?

As the reviewer correctly assumes, age groups were included as a factor in the ANOVA. We now state this more explicitly in the methods section (l. 488). The reference to the Supplementary Figure and Notes were corrected.

#4.4

Why not consider including all the age groups in the model as categorical variable rather than making pair-wise comparison. I understand you see the most differences between the youngest and the oldest groups but you are losing power.

We thank the reviewer for this excellent suggestion. Please see also our response #1.1 to reviewer 1. For the process analysis, we considered the young and the two old age groups

as a factor in the ANOVA (i.e. this is not based on pairwise comparisons). As suggested, we repeated our analysis while including data from all age groups for Gene Ontology. We found a total of 333 processes with a FDR p -value < 0.1 in this analysis (out of 523 processes that pass the model assumptions of the ANOVA). For the comparison between the young and the two old age groups we observe 448 processes with a FDR p -value < 0.1 (out of 900 processes that passed the model assumptions). Among the top 50 age-regulated processes across all age groups, 49 are also age-regulated in the comparison between the young and the two old age groups (87 among the top 100). Thus, considering all age groups seems not to increase the statistical power of the analysis likely due to the larger number of processes that failed the model assumptions. Please note that the above numbers of ageing regulated processes are not filtered according to our criteria that significant regulation across species additionally requires significant regulation in at least one mammalian and one fish species in the same direction (since we cannot precisely define up- or down-regulation if five age groups are considered). The corresponding information is now provided in the methods section (l. 520-525) and Supplementary Note S8.2.

#4.5

Can you indicate how many processes were tested and how many failed the model assumption tests?

For Gene Ontology 900 out of 1563, for KEGG Pathways 74 out of 135 and for Human Metabolic Pathways 43 out of 66 processes passed the model assumption tests. This information is now mentioned in the methods section (l. 494-496).

#4.6

Comparison of aging and disease signature:

What are the age ranges of "old individuals" the data from where the AMDA is calculated? I am encouraged that the AMDA for cancer in middle age is positive. Is there an AMDA for the middle age group for the chronic diseases? I'd expect at the younger age group the score would be negative then positive from middle age onward?

For the "old individuals" we used differentially expressed genes between the young and the two old age groups. This information is now provided in the methods section (l. 538). As detailed in Supplementary Note S5 we observed a positive AMDA in the comparison between the age groups 30 to 40 vs. 20 to 30 both for monocyte and T-cell data. Concerning degenerative diseases, we observe mostly negative AMDA scores for T-cells (4 positive, 9 negative) while we observe more positive scores for monocytes (5 positive, 3 negative). Thus, it is difficult to judge about the specific effect since there might also be a gender bias as discussed in Supplementary Note S5.

#4.7

Longitudinal analysis of aging cohorts

How relevant would disease expression scores from human tissue samples be for non-humans particularly when the aging genes across species lack consistency as you discuss in the paper?

In our manuscript, we indicated that previous studies did not find consistent ageing-regulated genes across species. We have now included additional data in which we compared ageing

regulated gene sets between all tissues and species also including previously published human data (l. 141-142 and Supplementary Note S3). We found a strong concordance in ageing-regulated gene sets in the pairwise comparisons with only few exceptions. With respect to the question whether disease-associated expression changes in humans reflect the corresponding expression changes in animal models our observations on associations between ageing-associated gene expression changes across non-human species with human disease data sets strongly supports that they are indeed representative.

#4.8

Shared genetic risk allele

This is an interesting idea and certainly could be part of the reason for the lack of association between disease SNP and longevity. Generally for shared risk alleles, were the effects sizes comparable for malignant and non-malignant SNPs? I do think that there are many other reasons including the relatively small effect sizes and/or variance explained by individual SNPs.

We thank the reviewer for this comment. We have toned down the corresponding statements (l. 103-105, l. 273-274, l. 348-349) and more specifically state that this antagonism might contribute to the lack of more associations. Moreover, we have checked effect sizes based on the odds ratios provided in the corresponding GWAS studies. The reported odds ratios for degenerative diseases are in the order of 1.24 (+/- 0.22) and 1.19 (+/- 0.09) for cancers and hence in a similar range. The corresponding data is now provided in the Supplementary Note S8.4 and Supplementary Data S1 along with the reported disease associations.

References

1. Deelen, J. *et al.* Genome-wide association meta-analysis of human longevity identifies a novel locus conferring survival beyond 90 years of age. *Hum Mol Genet* **23**, 4420-32 (2014).
2. Baumgart, M. *et al.* Longitudinal RNA-Seq Analysis of Vertebrate Aging Identifies Mitochondrial Complex I as a Small-Molecule-Sensitive Modifier of Lifespan. *Cell Syst* **2**, 122-32 (2016).
3. Wellcome Trust Case Control, C. Genome-wide association study of 14,000 cases of seven common diseases and 3,000 shared controls. *Nature* **447**, 661-78 (2007).
4. Harismendy, O. *et al.* 9p21 DNA variants associated with coronary artery disease impair interferon-gamma signalling response. *Nature* **470**, 264-8 (2011).
5. Almontashiri, N.A. *et al.* Interferon-gamma activates expression of p15 and p16 regardless of 9p21.3 coronary artery disease risk genotype. *J Am Coll Cardiol* **61**, 143-7 (2013).
6. Almontashiri, N.A. The 9p21. 3 risk locus for coronary artery disease: A 10-year search for its mechanism. *Journal of Taibah University Medical Sciences* (2017).
7. Cellerino, A., Valenzano, D.R. & Reichard, M. From the bush to the bench: the annual *Nothobranchius* fishes as a new model system in biology. *Biol Rev Camb Philos Soc* **91**, 511-33 (2016).
8. Berger, N.A. *et al.* Cancer in the elderly. *Trans Am Clin Climatol Assoc* **117**, 147-55; discussion 155-6 (2006).
9. de Magalhaes, J.P. How ageing processes influence cancer. *Nat Rev Cancer* **13**, 357-65 (2013).

2.6 The authors have clarified the relation between somatic copy number variations derived from tumors in aging-induced and aging-suppressed genes.

2.7 The remark on the contribution of the *Nothobranchius furzeri* dataset has been properly addressed by explicitly mentioning this in the main text (l.120).

2.8 The authors have identified a list of 896 genes with conserved regulation in multiple species and tissues. It is satisfactory and reconfirming that within this gene set down-regulated genes were found to be highly enriched for cell cycle-associated processes, while up-regulated genes appeared enriched for immune-associated processes.

Other minor remarks were properly addressed.

Overall: With the exception of part of point 2.3 this referee is happy with the manner the authors have responded to the comments. This has led to substantial improvements of the manuscript.

Reviewer #3 (Remarks to the Author):

I feel the authors worked hard to adequately address most comments and I have no additional major comments. I do suggest the authors in future may want to consider further ways to explore the effect of bias in cancer gene expression tumor tissues vs. mostly blood-based degenerative disease expression datasets. I do not think this comment was fully addressed though given available datasets it is difficult to address.

Reviewer #4 (Remarks to the Author):

All my comments have been adequately addressed.